# Fission yeast Srr1 and Skb1 promote isochromosome formation at the centromere

Piyusha Mongia [1,2,6], Naoko Toyofuku[1,6], Ziyi Pan [1,2,6], Ran Xu[1,2], Yakumo Kinoshita[1,2], Keitaro Oki[1], Hiroki Takahashi[3], Yoshitoshi Ogura[4], Tetsuya Hayashi [5] & Takuro Nakagawa [1,2✉]

Rad51 maintains genome integrity, whereas Rad52 causes non-canonical homologous recombination leading to gross chromosomal rearrangements (GCRs). Here we find that fission yeast Srr1/Ber1 and Skb1/PRMT5 promote GCRs at centromeres. Genetic and physical analyses show that *srr1* and *skb1* mutations reduce isochromosome formation mediated by centromere inverted repeats. *srr1* increases DNA damage sensitivity in *rad51* cells but does not abolish checkpoint response, suggesting that Srr1 promotes Rad51-independent DNA repair. *srr1* and *rad52* additively, while *skb1* and *rad52* epistatically reduce GCRs. Unlike *srr1* or *rad52*, *skb1* does not increase damage sensitivity. Skb1 regulates cell morphology and cell cycle with Slf1 and Pom1, respectively, but neither Slf1 nor Pom1 causes GCRs. Mutating conserved residues in the arginine methyltransferase domain of Skb1 greatly reduces GCRs. These results suggest that, through arginine methylation, Skb1 forms aberrant DNA structures leading to Rad52-dependent GCRs. This study has uncovered roles for Srr1 and Skb1 in GCRs at centromeres.

[1] Department of Biological Sciences, Graduate School of Science, Osaka University, 1—1 Machikaneyama, Toyonaka, Osaka 560-0043, Japan. [2] Forefront Research Center, Graduate School of Science, Osaka University, 1—1 Machikaneyama, Toyonaka, Osaka 560-0043, Japan. [3] Medical Mycology Research Center, Chiba University, Chiba 260-8673, Japan. [4] Division of Microbiology, Department of Infectious Medicine, Kurume University School of Medicine, Kurume, Fukuoka 830-0011, Japan. [5] Department of Bacteriology, Faculty of Medical Sciences, Kyushu University, Fukuoka 812-8582, Japan. [6] These authors contributed equally: Piyusha Mongia, Naoko Toyofuku, Ziyi Pan. ✉email: nakagawa.takuro.sci@osaka-u.ac.jp

Gross chromosomal rearrangements (GCRs), such as translocations, can occur using repetitive sequences that are abundant and widespread in eukaryotic genomes[1]. In humans, the total number of repetitive sequences, including satellite repeats and transposable elements, accounts for 54% of the genome[2,3]. GCRs cause cell death and genetic disorders, including cancer. On the other hand, GCRs can be a driving force of evolution by creating genome diversity[4]. Therefore, GCRs are not only pathological but also physiological phenomena.

The centromere that ensures proper chromosome segregation contains repetitive DNA sequences in many eukaryotes. Human centromeres (≥ 3 Mb) contain α satellite and other types of satellite repeats, transposable elements, and segmental duplications[5]. The orientation of the centromere repeats including higher-order repeats of α satellites, switches within a centromere, forming inverted DNA repeats. Despite the important role in chromosome segregation, the centromere is a hotspot for chromosomal breakage and rearrangement[6–10]. Recombination between repetitive sequences at the centromere forms abnormal chromosomes[11–13]. Robertsonian translocations, a fusion of two acrocentric chromosomes at or around centromeres, are the most frequently observed form of chromosomal abnormality in humans, affecting 1 out of 1000 newborns[14]. Isochromosomes whose arms are mirror images of each other are commonly found in cancer cells[15]. Isochromosomes of chr21 and chrX cause Down and Turner syndromes, respectively[16,17]. Compared to mammalian centromeres, the fission yeast *S. pombe* centromeres are short (35~110 kb) but contain inverted DNA repeats flanking a non-repetitive core sequence[18,19]. In this fungus, isochromosomes are produced using inverted DNA repeats in the centromere[20–22]. Less complexity of the centromere DNA sequence makes fission yeast an excellent system to study the mechanism of centromeric GCRs.

Homologous recombination is required to repair detrimental DNA damage such as double-strand breaks[23]. Rad51 is the key player in canonical homologous recombination and catalyses homology search and DNA strand exchange, forming displacement loops. Mammalian BRCA1 and BRCA2 facilitate Rad51-dependent recombination, and their mutations increase GCRs and predispose the carriers to cancer[24,25]. Homologous recombination maintains centromere integrity. In mammals, the inactivation of Rad51 increases aberrant recombination at centromeres[9,10,26]. In fission yeast, loss of Rad51 increases isochromosome formation at centromeres[20,21,27]. Detailed analysis using fission yeast showed that Rad51 preferentially promotes a conservative way of recombination: non-crossover recombination at centromeres[27,28], thereby suppressing isochromosome formation.

Another recombinase Rad52 promotes homology-dependent DNA recombination/repair independent of Rad51[29,30]. Rad52, on its own, promotes displacement loop formation, single-strand annealing (SSA), and inverse-strand exchange using RNA strands. Yeast Rad52 also facilitates Rad51 loading onto replication protein A (RPA)-coated single-stranded DNA, while human Rad52 does not have the loader activity[31]. In both mammals and fission yeast, Rad52-dependent non-canonical recombination causes GCRs at centromeres[9,32]. In fission yeast, Rad52 causes isochromosome formation via crossover recombination with Mus81, a crossover-specific endonuclease[27,32–35]. PCNA ubiquitination at lysine 107 and Msh2-Msh3 have been implicated in the Rad52-dependent GCR pathway[32,36]. The DNA sliding clamp PCNA may form DNA structures leading to Rad52-dependent GCRs because PCNA K107 is dispensable for DNA damage repair[36]. The rad52 deletion does not eliminate isochromosome formation, suggesting the presence of a Rad52-independent GCR pathway(s). Moreover, the initial event that leads to GCRs remains unclear.

To gain insights into the GCR mechanism, we search for the factors that cause GCRs in the *rad51Δ* mutant strain and find Srr1 and Skb1. In *A. thaliana* and mice, the Srr1 homolog affects the transcription of the genes involved in the circadian rhythm[37–39]. Skb1 is involved in a range of pathways, including cell morphology and cell cycle regulation in fission yeast[40–43], and is the homolog of the human protein arginine methyltransferase 5 (PRMT5)[44,45]. Srr1 and Skb1 specifically promote isochromosome formation. Remarkably, the *srr1* mutation increases DNA damage sensitivity and chromosome loss but is not essential for checkpoint response to DNA damage, suggesting that Srr1 promotes DNA damage repair. *srr1* and *rad52* mutations additively reduced GCR rates, suggesting that Srr1 and Rad52 have overlapping and non-overlapping roles in GCRs. In contrast to *srr1*, the *skb1* deletion does not increase DNA damage sensitivity and, intriguingly, reduces chromosome loss in *rad51Δ* cells. Loss of Slf1[40,41] or Pom1[42,43], which functions with Skb1 in cell morphology and cell cycle regulation, did not reduce GCRs. However, mutating conserved residues in the arginine methyltransferase (RMTase) domain of Skb1 strongly reduced GCRs, suggesting that Skb1 causes isochromosome formation through its RMTase activity. These findings pave new avenues to decipher the mechanism of GCR events at the centromere.

## Results

**Srr1 and Skb1 cause gross chromosomal rearrangements (GCRs).** To gain insights into the mechanism of GCRs, we introduced random mutations into *rad51Δ* cells that show elevated GCR rates and searched for the clones that exhibit reduced levels of GCRs. To detect otherwise lethal GCRs in haploid cells, we used an extra-chromosome ChL$^C$ (~530 kb) derived from fission yeast chromosome 3 (chr3) and detected spontaneous GCRs that had lost *ura4$^+$* and *ade6$^+$* marker genes (Fig. 1a)[20,27,46]. To assess GCR rates, yeast clones grown on Edinburgh minimum media supplemented with uracil and adenine (EMM+UA) were replicated onto the media containing 5-fluoroorotic acid (5-FOA+UA) that is toxic to *ura4$^+$* cells. Of 24,000 clones, three reproducibly exhibited reduced levels of GCRs. Genome sequencing of one of them identified the *srr1/ber1-W157R* and *skb1-A377V* mutations in their SRR1-like and arginine methyltransferase (RMTase) domains, respectively (Fig. 1b). The *srr1* and *skb1* genes are only 51 kb apart from each other on chr2. The replica plating assay shows that, compared to the parental *rad51Δ* strain, the *rad51Δ* clone containing the *srr1* and *skb1* mutations from the reduced number of colonies on the 5-FOA+UA plate (Fig. 1c).

To establish whether Srr1 or Skb1 is required for GCRs, we deleted the genes and determined GCR rates by the fluctuation test (Fig. 1d). In the wild-type background, *srr1Δ* but not *skb1Δ* slightly reduced GCR rates, showing that Srr1 is required for GCRs even in the presence of Rad51. To our surprise, not only *srr1Δ* but also *skb1Δ* reduced GCR rates in the *rad51Δ* background, demonstrating that both Srr1 and Skb1 cause GCRs. Remarkably, *srr1Δ skb1Δ* double mutation further reduced GCRs than the single mutations, suggesting that Srr1 and Skb1 have non-overlapping roles in GCRs (see below). To determine whether the *srr1-W157R* and *skb1-A377V* point mutations reduce GCR rates, we introduced each mutation into yeast by transformation (see Methods) and determined GCR rates in the *rad51Δ* background (Fig. 1e). *srr1-W157R* reduced GCR rates although less prominent than *srr1Δ*, suggesting a residual activity of the Srr1 mutant protein. Unlike *skb1Δ*, *skb1-A377V* did not significantly reduce GCR rates in *srr1$^+$ rad51Δ* cells. However, *skb1-A377V* reduced GCR rates in *srr1-W157R rad51Δ* cells, suggesting that *skb1-A377V* partially inactivates the Skb1 function. The additive effect of *srr1-W157R* and *skb1-A377V* on GCR rates can explain why our genetic screening identified both

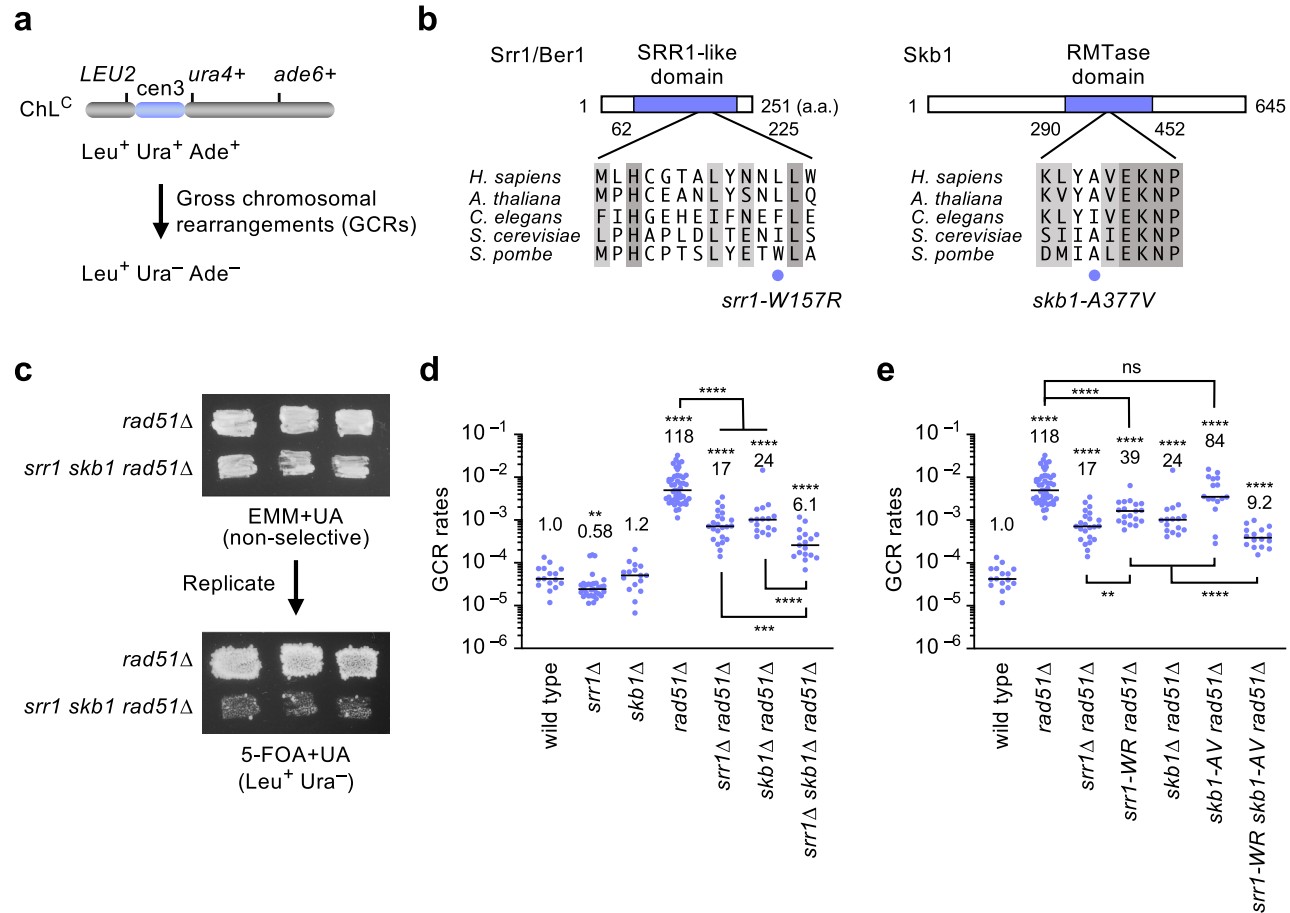

**Fig. 1 Srr1 and Skb1 promote GCRs in fission yeast. a** Depicted is an extra-chromosome ChL$^C$ used to detect GCRs in this study. GCRs resulting in Leu$^+$ Ura$^-$ Ade$^-$ from Leu$^+$ Ura$^+$ Ade$^+$ were detected. **b** Srr1/Ber1 and Skb1 proteins contain the SRR1-like domain and the arginine methyltransferase (RMTase) domain, respectively. Aligned are the amino acid sequences around the *srr1-W157R* and *skb1-A377V* mutations (blue circles) in different species. Similar and identical residues among the different species are highlighted in pale and dark gray, respectively. **c** The parental *rad51Δ* strain and the clone additionally containing *srr1-W157R* and *skb1-A377V* mutations (TNF5411 and 5954) grown on EMM+UA were replicated onto 5-FOA+UA plates. Leu$^+$ Ura$^-$ cells selectively form colonies on 5-FOA+UA plates. **d** GCR rates of wild-type, *srr1Δ*, *skb1Δ*, *rad51Δ*, *srr1Δ rad51Δ*, *skb1Δ rad51Δ*, and *srr1Δ skb1Δ rad51Δ* strains (TNF5369, 5774, 5772, 5411, 5904, 5788, and 8432). **e** GCR rates of wild-type, *rad51Δ*, *srr1Δ rad51Δ*, *srr1-W157R rad51Δ* (TNF8344), *skb1Δ rad51Δ*, *skb1-A377V rad51Δ* (TNF8359), and *srr1-W157R skb1-A377V rad51Δ* (TNF8547). Each dot represents a value obtained from an independent experiment. Black lines show the median. Rates relative to the wild-type are shown on top of each dot cluster. The two-tailed Mann-Whitney test between the wild-type and mutant strains and between the indicated pairs. ns, non-significant; **$p < 0.01$; ***$p < 0.001$; ****, $p < 0.0001$. Numerical data underlying the graphs **d** and **e** are provided in Table A in Supplementary Data 1.

mutations in the same clone. These results are consistent with the idea that Srr1 and Skb1 have non-overlapping roles in GCRs.

Budding yeast *srr1/ber1Δ* (benomyl resistant) cells are hyper-resistant to microtubule-destabilizing benzimidazole compounds: benomyl and nocodazole[39]. However, fission yeast *srr1Δ* cells were no more resistant than wild type to thiabendazole (TBZ), the most popular benzimidazole used in this fungus (Supplementary Fig. 1). Based on this, we decided to call this gene *srr1* rather than *ber1* to avoid confusion.

**Srr1 and Skb1 promote isochromosome formation at the centromere.** Previous studies showed that *rad51Δ* cells produce isochromosomes and relatively few chromosomal truncations[27,32,36]. Isochromosomes are produced using inverted DNA repeats at the centromere, whereas chromosomal truncations are formed by the de novo addition of telomere sequences to new chromosome ends (Fig. 2a). Isochromosomes (300~400 kb) and truncations (< 220 kb) are distinguished from each other by their lengths. To determine which GCRs Srr1 and Skb1 cause, chromosomal DNAs of the parental and independent GCR clones were prepared in agarose plugs,

separated by pulse-field gel electrophoresis (PFGE), and stained with ethidium bromide (Fig. 2b). The sizes of GCR products were determined using the parental ChL$^C$ (530 kb) and the lambda (λ) DNA ladder as references. *rad51Δ* cells produced many iso-chromosomes (300~400 kb) and a small number of truncations (< 220 kb), as observed previously[20,27,32,36]. Loss of Srr1 or Skb1 reduced the proportion of isochromosomes among GCR products. The total GCR rates were multiplied by the proportion of iso-chromosomes or truncations (Table 1) to obtain the rates of iso-chromosomes and truncations, respectively (Fig. 2c). Either *srr1Δ* or *skb1Δ* eliminated ~90% of isochromosomes formed in *rad51Δ* cells, but neither reduced chromosomal truncations. These data indicate that Srr1 and Skb1 are specifically required for isochromosome formation at the centromere.

**Srr1 has a Rad51-independent role in DNA damage repair and chromosome maintenance.** As Srr1 and Skb1 promote iso-chromosome formation mediated by centromere repeats, they might be involved in the recombinational repair of DNA damage. To test this possibility, we performed a serial dilution assay and

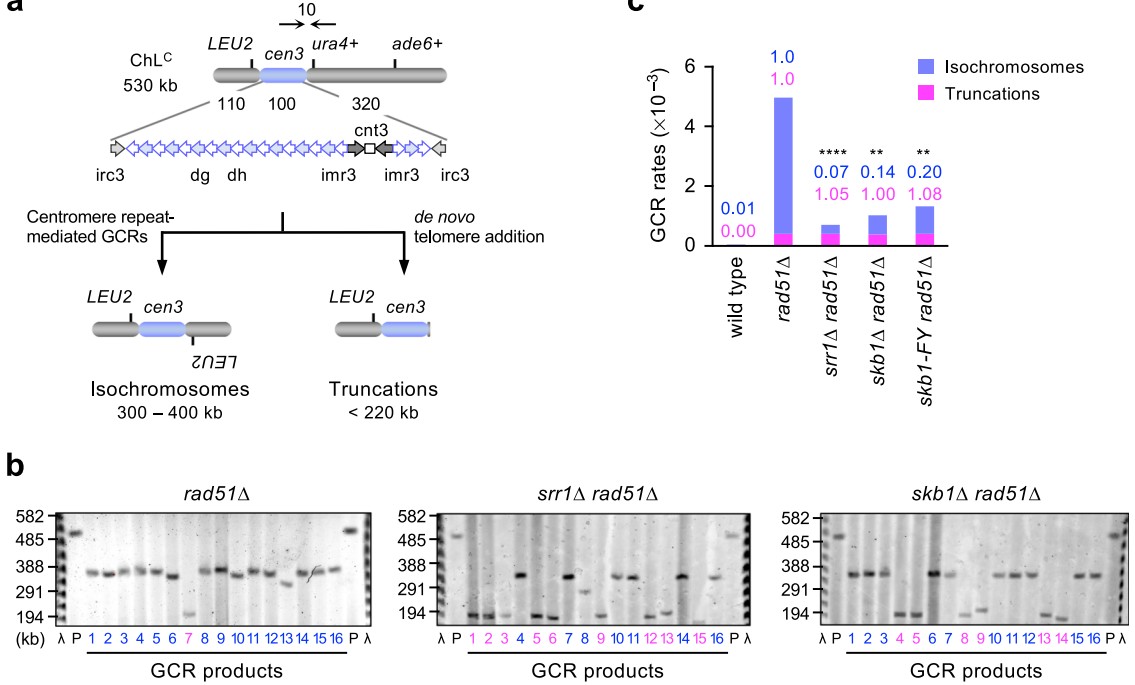

**Fig. 2 Srr1 and Skb1 cause isochromosome formation but not chromosomal truncation. a** Depicted are the non-repetitive (cnt3) and repetitive sequences (the innermost imr3, dg, dh, and the outermost irc3) in cen3. The ura4+ marker gene is placed at 10 kb from cen3. Loss of Rad51 increases isochromosomes and chromosomal truncations. **b** Chromosomal DNAs prepared from the parental (P) and independent GCR clones of rad51Δ, srr1Δ rad51Δ, and skb1Δ rad51Δ strains (TNF5411, 5904, and 5788) were separated by PFGE. Sizes of lambda (λ) DNA ladders are indicated on the left of the panels. Sample numbers of isochromosomes and truncations are shown in blue and magenta, respectively. **c** Rates of isochromosome formation and chromosomal truncation in wild-type (TNF5369), rad51Δ, srr1Δ rad51Δ, skb1Δ rad51Δ, and skb1-F319Y rad51Δ (TNF8391) strains. Rates relative to the rad51Δ strain are indicated on the top of the bars. The two-tailed Fischer's exact test between the rad51Δ and other mutant strains. **p < 0.01; ****p < 0.0001. Numerical data underlying **c** are provided in Table B in Supplementary Data 1. Uncropped gel images are shown in Supplementary Fig. 5.

**Table 1 Srr1 and Skb1 are specifically required for isochromosome formation.**

| | Strain # | Translocations | Isochromosomes | Truncations | Total |
|---|---|---|---|---|---|
| wild type | TNF5369 | 1 (3%) | 31 (97%) | 0 (0%) | 32 |
| rad51Δ | TNF5411 | 0 (0%) | 72 (92%) | 6 (8%) | 78 |
| srr1Δ rad51Δ | TNF5904 | 0 (0%) | 7 (44%) | 9 (56%) | 16 |
| skb1Δ rad51Δ | TNF5788 | 0 (0%) | 10 (63%) | 6 (37%) | 16 |
| skb1-FY rad51Δ | TNF8391 | 0 (0%) | 22 (69%) | 10 (31%) | 32 |

The percentage of each GCR type is shown in the parenthesis. GCR products of wild type have been shown previously[36]. Some GCR products of rad51Δ have been shown previously[32, 36].

determined the sensitivity of srr1 and skb1 mutant strains to DNA-damaging agents (Fig. 3a). Methyl methanesulfonate (MMS) is a DNA alkylating agent; hydroxyurea (HU) depletes dNTP pool; camptothecin (CPT) is a topoisomerase inhibitor. These agents interfere with the progression of replication forks and create DNA breaks. Compared to wild-type, srr1Δ cells exhibited hypersensitivity to all the DNA-damaging agents (Fig. 3a, top panels). Notably, srr1Δ rad51Δ cells were more sensitive than the single mutants, suggesting a role for Srr1 in Rad51-independent DNA damage response. The srr1-W157R mutation that partially reduced GCR rates (Fig. 1e) also partially increased the damage sensitivity. These results suggest that Srr1 facilitates Rad51-independent DNA damage response.

Cell cycle arrest caused by DNA damage checkpoint gives cells time to repair DNA. To determine whether Srr1 is required to arrest cell cycle progression in response to DNA damage, we determined the percentage of septated cells before and after exposure to MMS (Fig. 3b). In the wild-type, the septation index declined from ~30 to 5% after MMS exposure, demonstrating

MMS-induced cell cycle arrest. The checkpoint kinase Chk1 is required for cell cycle arrest[47]. In the chk1Δ strain, the septation index did not decline to the wild-type level. Unlike chk1Δ, in the srr1Δ strain, the septation index declined to the wild-type level by 6 h after MMS addition, suggesting that Srr1 is dispensable for cell cycle arrest. A delay in the reduction of septated cells can be due to the slow growth of the srr1Δ strain. The doubling time of wild-type and srr1Δ cells grown in EMM media at 30 °C were 2.48 ± 0.11 and 2.73 ± 0.10 h, respectively (p = 0.042, the two-tailed student's t-test) (Table E in Supplementary Data 1). Like wild-type cells, srr1Δ cells were elongated after MMS exposure (Supplementary Fig. 2a). These data suggest that Srr1 is dispensable for DNA damage-induced cell cycle arrest. Chk1 kinase is phosphorylated and activated in response to DNA damage[48,49]. To determine whether Srr1 is required for the Chk1 phosphorylation, HA-tagged Chk1-HA was expressed from the original chromosomal locus, separated by SDS-PAGE, and detected using anti-HA antibodies (Fig. 3c). A slow-migrating Chk1-HA band indicative of the phosphorylation[48] was observed upon MMS treatment in both wild-type and srr1Δ

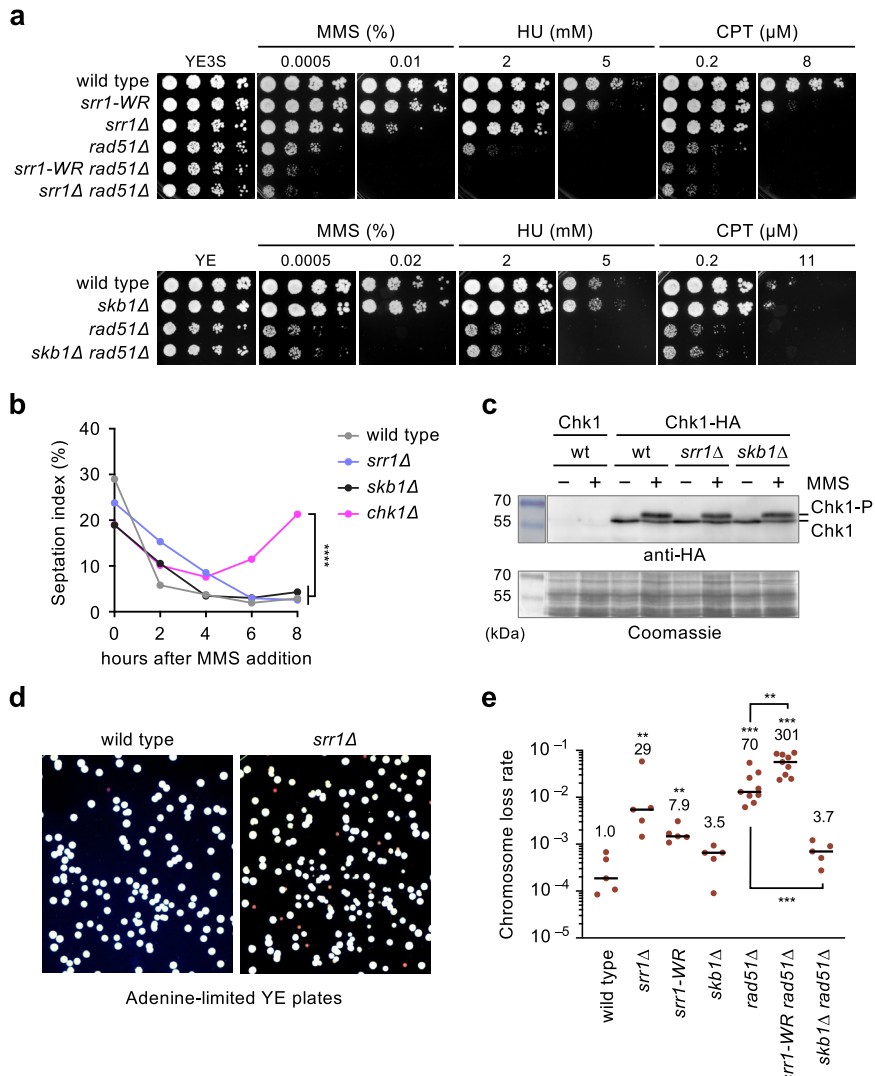

**Fig. 3 *srr1* but not *skb1* mutations increase DNA damage sensitivity. a** A serial dilution assay to determine the sensitivity to DNA damaging agents. Log-phase cultures of wild-type, *srr1-W157R, srr1Δ, rad51Δ, srr1-W157R rad51Δ*, and *srr1Δ rad51Δ* (TNF3885, 8280, 5847, 5845, 8573, and 5849) prepared in YE3S medium were spotted onto YE3S plates supplemented with the indicated concentrations of MMS, HU, or CPT (top panels). Log-phase cultures of wild-type, *skb1Δ, rad51Δ*, and *skb1Δ rad51Δ* (TNF35, 8321, 8107, and 8320) prepared in YE medium were spotted onto YE plates (bottom panels). **b** Wild-type, *srr1Δ, skb1Δ*, and *chk1Δ* cells (TNF35, 5943, 8321, and 3559) in the log phase in EMM were treated with 0.01% MMS. The percentage of cells containing a septum is indicated. > 300 cells are counted at each point. Pearson's Chi-square test of the septation index between wild-type and other strains at $t = 8$ h showed that *srr1Δ* or *skb1Δ* did not significantly change the septation index ($p > 0.05$) but *chk1Δ* increased it. **c** Chk1 phosphorylation in response to MMS treatment. Before and after 4 h treatment with 0.01% MMS, extracts were prepared from *chk1+* (TNF7555) and *chk1-HA+* cells of wild-type, *srr1Δ*, and *skb1Δ* (TNF8441, 8799, and 8802) and separated by 8% SDS-PAGE. Chk1-HA was detected by Western blotting using anti-HA antibodies (16B12). Whole proteins were stained using Coomassie brilliant blue. Size markers (Takara, 3454 A, CLEARLY stained protein ladder) are shown on the left. wt, wild-type. Uncropped images are shown in Supplementary Fig 6. **d** Wild-type and *srr1Δ* (TNF5369 and 5774) cells were plated onto adenine-limited YE plates, on which *ade6−* cells form red colonies. **e** Chromosome loss rates of wild-type, *srr1Δ, srr1-W157R, skb1Δ, rad51Δ, srr1-W157R rad51Δ*, and *skb1Δ rad51Δ* strains (TNF5369, 5774, 8308, 5772, 5411, 8344, and 5788). The two-tailed Mann-Whitney test. **$p < 0.01$; ***$p < 0.001$. Numerical data underlying **b**, **e** are provided in Tables C and D, respectively, in Supplementary Data 1.

strains, showing that Srr1 is not essential for DNA damage checkpoint activation.

Repair of spontaneous DNA damage is vital to maintaining chromosomes. To determine whether Srr1 is required to maintain chromosomes, we determined spontaneous loss rates of ChL$^C$ by the fluctuation test. A colony formed on YE3S plates was suspended in sterilized water, and the cells were plated onto adenine-limited YE plates, on which cells lacking *ade6+* formed red colonies[50] (Fig. 3d). The red colonies were further inspected for Leu and Ura auxotrophy to obtain the rate of chromosome loss (i.e., Leu− Ura− Ade−) (Fig. 3e). We found that *srr1Δ* and *srr1-W157R* increased the rate of

chromosome loss. *rad51Δ* also increases chromosome loss, as observed previously[32]. Notably, *srr1-W157R* and *rad51Δ* synergistically increased chromosome loss, demonstrating that Srr1 and Rad51 have different roles in maintaining chromosomes.

In contrast to the *srr1* mutations, *skb1Δ* did not increase the sensitivity to DNA-damaging agents either in the presence or absence of Rad51 (Fig. 3a, bottom panels), showing that Skb1 is dispensable to repair DNA damage induced by MMS, HU, or CPT. Skb1 was also dispensable for cell cycle arrest and Chk1 phosphorylation induced by MMS treatment (Fig. 3b, c). Intriguingly, however, *skb1Δ* strongly reduced chromosome loss

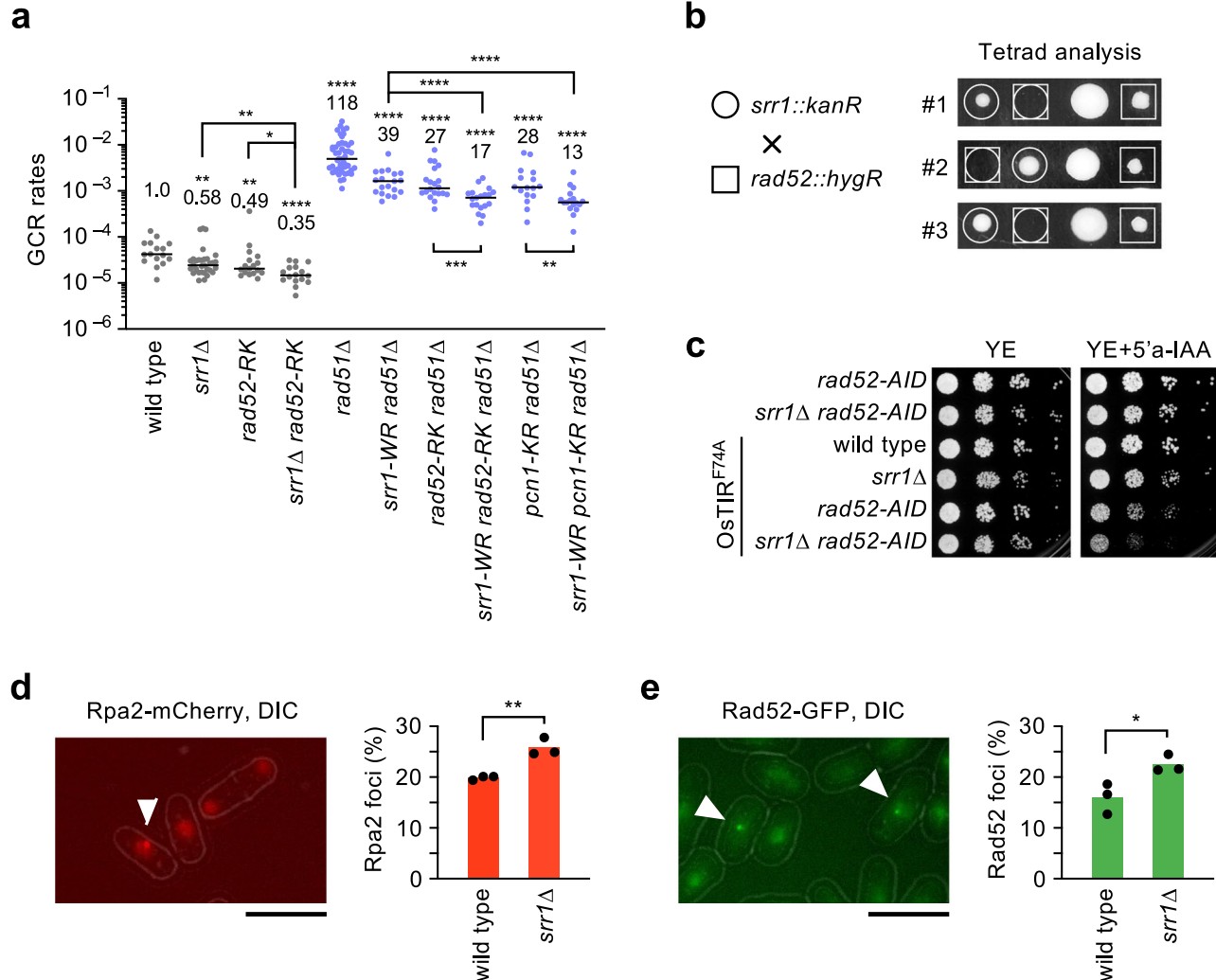

**Fig. 4 Srr1 plays a role in the Rad52-independent GCR pathway. a** GCR rates of wild-type, *srr1Δ*, *rad52-R45K*, *srr1Δ rad52-R45K*, *rad51Δ*, *srr1-W157R rad51Δ*, *rad52-R45K rad51Δ*, *srr1-W157R rad52-R45K rad51Δ*, *pcn1-K107R rad51Δ*, and *srr1-W157R pcn1-K107R rad51Δ* strains (TNF5369, 5774, 6599, 8281, 5411, 8344, 7122, 8663, 6761, and 8601). The two-tailed Mann-Whitney test. **b** Tetrad analysis of *srr1Δ* and *rad52Δ*. *srr1::kanR* and *rad52::hygR* haploids (TNF5943 and 7988) were crossed, and the resulting tetrads were dissected on YE plates under a microscope. Images of three sets of three-spore viable tetrads in which the *srr1::kanR rad52::hygR* progenies did not form colonies are shown. **c** Depletion of Rad52 by the AID system impairs the growth of *srr1Δ* cells. *rad52-AID*, *srr1Δ rad52-AID*, *OsTIR^{F74A}*, *srr1Δ OsTIR^{F74A}*, *rad52-AID OsTIR^{F74A}*, and *srr1Δ rad52-AID OsTIR^{F74A}* (TNF8614, 8621, 8616, 8623, 8617, and 8627) were spotted on YE plates supplemented with 200 nM 5'a-IAA which induces Rad52 depletion. **d** Rpa2-mCherry foci (arrowhead) were observed by fluorescence microscopy in wild-type cells (TNF5492). Fluorescence and DIC images are overlayed. DIC, differential interference contrast. A bar shown below the image indicates 10 μm. The bar graph shows percentages of nuclei containing at least one Rpa2-mCherry focus in wild-type and *srr1Δ* (TNF8803) strains. The bars represent the mean of three independent experiments. The two-tailed student's *t*-test. **e** Rad52-GFP foci (arrowhead) were observed in wild-type cells (TNF4442). The bar graph shows percentages of nuclei containing at least one Rad52-GFP focus in wild-type and *srr1Δ* (TNF6130) strains. Numerical data underlying **a** are provided in Table A, and those underlying **d** and **e** are in Table F in Supplementary Data 1.

in *rad51Δ* cells (Fig. 3e), showing that Skb1 causes isochromosome formation and chromosome loss in *rad51Δ* cells.

**Srr1 and Rad52 have overlapping and non-overlapping roles in GCRs.** Rad52 promotes isochromosome formation in *rad51Δ* cells. The *rad52-R45K* mutation in the N-terminal DNA-binding domain specifically impairs in vitro single-strand annealing (SSA) activity and reduces isochromosome formation to the same extent as *rad52Δ*[32], suggesting a role of Rad52-mediated SSA in isochromosome formation. The *rad52-R45K*, *rad52Δ*, and *srr1Δ* mutations eliminate ~90% of isochromosomes in *rad51Δ* cells (Fig. 2c and ref. [32]), indicating that both Rad52 and Srr1 are essential for the major pathway of isochromosome formation. To

define the relationship between Srr1 and Rad52, we created the *srr1 rad52* double mutants and determined GCR rates (Fig. 4a). Notably, *srr1Δ* and *rad52-R45K* additively reduced GCR rates in the wild-type background, suggesting that Srr1 and Rad52 also have non-overlapping roles in GCRs. Consistent with this idea, *srr1-W157R* and *rad52-R45K* additively reduced GCR rates in the *rad51Δ* background. Ubiquitination of PCNA (encoded by the *pcn1* gene) at lysine 107 (K107) plays a role in Rad52-dependent GCRs[36]. As expected, *srr1-W157R* and *pcn1-K107R* also additively reduced GCR rates in *rad51Δ* cells (Fig. 4a). These data show that Srr1 and Rad52 have both overlapping and non-overlapping roles in GCRs.

We crossed *srr1Δ* and *rad52Δ* haploid strains and dissected the tetrads but failed to obtain *srr1Δ rad52Δ* progenies (Fig. 4b),

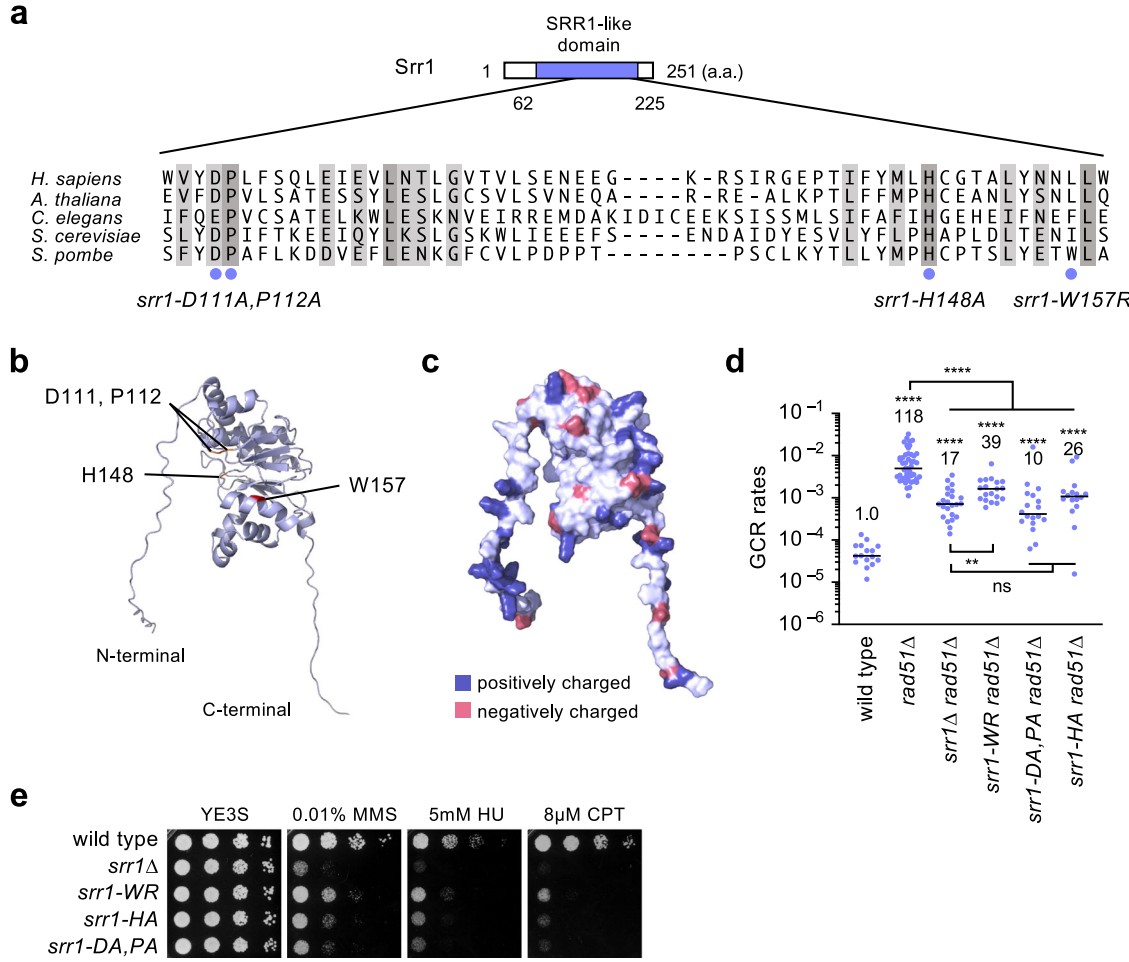

**Fig. 5 A role of the SRR1-like domain in GCRs and DNA damage repair. a** Positions of the fission yeast *srr1-D111A,P112A*, *-H148A*, and *-W157R* mutation sites are indicated by blue circles. Similar and identical residues among the different species are highlighted in pale and dark gray, respectively. **b** A ribbon model of the Srr1 structure predicted by AlphaFold methods. Positions of the mutation sites are indicated. **c** A surface model of the Srr1 structure. Positively and negatively charged residues are shown in blue and red, respectively. **d** GCR rates of wild-type, *rad51Δ*, *srr1Δ rad51Δ*, *srr1-W157R rad51Δ*, *srr1-D111A,P112A rad51Δ*, and *srr1-H148A rad51Δ* (TNF5369, 5411, 5904, 8344, 8686, and 8387). The two-tailed Mann-Whitney test. Numerical data are provided in Table A in Supplementary Data 1. **e** Mutating the conserved residues in the SRR1-like domain increases the sensitivity to MMS, HU, and CPT. Wild-type, *srr1Δ*, *srr1-W157R*, *srr1-H148A*, and *srr1-D111A,P112A* (TNF3885, 5847, 8280, 8275, and 8274) cells were spotted onto YE3S supplemented with the indicated concentrations of MMS, HU, or CPT.

suggesting synthetic growth defects of the *srr1 rad52* double deletion. To confirm this, we depleted Rad52 in *srr1Δ* cells using the auxin-induced degron 2 (AID2) system[51]. In the presence of the F-box protein OsTIR1[F74A], an auxin analog 5'-adamantyl-IAA (5'a-IAA) induces polyubiquitin-dependent degradation of the AID-tagged Rad52 protein, Rad52-AID. The addition of 5'a-IAA to the media significantly impaired cell growth of the *srr1Δ rad52-AID* compared to the *rad52-AID* strain (Fig. 4c, last two rows). These results show non-overlapping roles for Srr1 and Rad52 in cell growth. Given the hypersensitivity of *srr1Δ* cells to DNA-damaging agents (Fig. 3a), Srr1 may promote the repair of spontaneous DNA damage. Single-stranded DNA is formed during replication, transcription, and DNA damage repair/recombination. The replication protein A (RPA) complex binds single-stranded DNA with high affinity[52]. We detected the spontaneous focus formation of Rpa2-mCherry expressed from the original chromosomal locus[32] and found that *srr1Δ* increased the fraction of cells containing Rpa2-mCherry focus (Fig. 4d), suggesting the accumulation of single-stranded DNA in *srr1Δ* cells. Rad52 forms nuclear foci at spontaneous DNA damage sites[27,53]. *srr1Δ* also increased the cells containing Rad52-GFP focus (Fig. 4e), showing that Srr1 suppresses the Rad52 focus formation.

**A role of the SRR1-like domain in GCRs and DNA damage response.** Srr1 has an evolutionarily conserved domain called the SRR1-like domain (Fig. 5a). The Srr1 protein structure predicted by AlphaFold methods[54,55] consists of the SRR1-like domain containing β-sheets sandwiched by α-helixes with intrinsically disordered extensions[56] at both N- and C-termini (Fig. 5b, c). The *srr1-W157R* mutation site is present in the SRR1-like domain. To extend this, we changed conserved residues in the SRR1-like domain to alanine: *srr1-D111A,P112A* and *srr1-H148A* (Fig. 5a, b). Both *srr1-D111A,P112A* and *srr1-H148A* mutations reduced GCR rates (Fig. 5d). *srr1-D111A,P112A* and *srr1-H148A* also increased DNA damage sensitivity (Fig. 5e). These results demonstrate that the SRR1-like domain plays an essential role in GCRs and DNA damage repair. As expected from the prolonged doubling time (Table E in Supplementary Data 1), *srr1Δ* cells formed small colonies on plate media compared to wild-type cells (Supplementary Fig. 2b). Like *srr1Δ* cells, *srr1-W157R*, *srr1-D111A,P112A*, and *srr1-H184A* cells produced small colonies (Supplementary Fig. 2b), consistent with the role of the SRR1-like domain even in the absence of exogenous DNA damage.

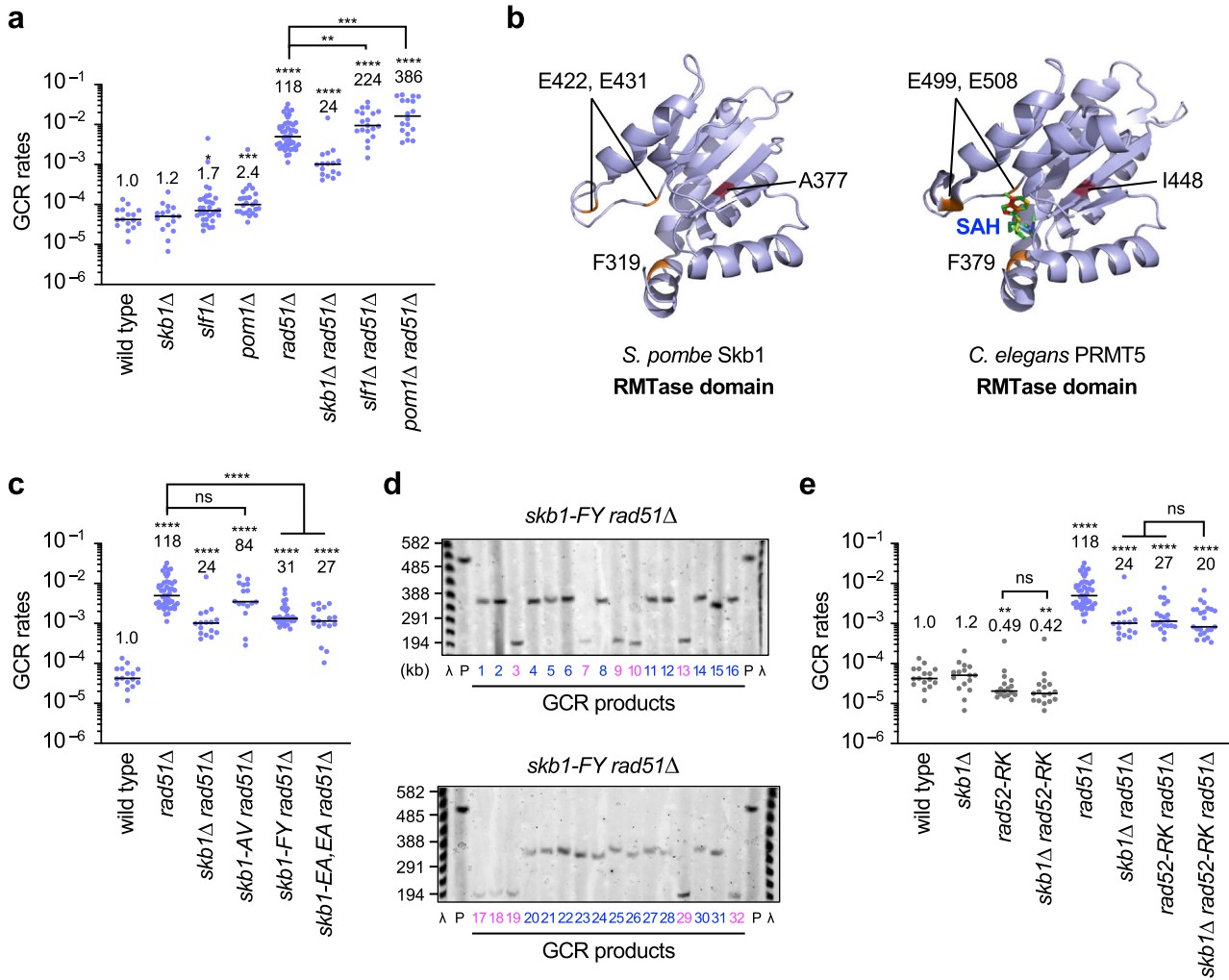

**Fig. 6 Skb1 arginine methyltransferase acts in the Rad52-dependent GCR pathway. a** GCR rates of wild-type, *skb1Δ*, *slf1Δ*, *pom1Δ*, *rad51Δ*, *skb1Δ rad51Δ*, *slf1Δ rad51Δ*, and *pom1Δ rad51Δ* strains (TNF5369, 5772, 8811, 8813, 5411, 5788, 8834, and 8838). **b** Shown are the structure of the arginine methyltransferase domain of the fission yeast *S. pombe* Skb1 predicted by AlphaFold methods and the crystal structure of the arginine methyltransferase domain of *C. elegans* PRMT5 (PDB code 3UA3) with SAH, a SAM analog. Positions of the phenylalanine (F) and glutamic acid (E) residues essential for arginine methyltransferase activity are indicated. **c** GCR rates of wild-type, *rad51Δ*, *skb1Δ rad51Δ*, *skb1-A377V rad51Δ*, *skb1-F319Y rad51Δ*, and *skb1-E422A,E431A rad51Δ* strains (TNF5369, 5411, 5788, 8359, 8391, and 8474). **d** PFGE separated GCR products of the *skb1-F319Y rad51Δ* strain. Sample numbers of isochromosomes and truncations are shown in blue and magenta, respectively. **e** GCR rates of wild-type, *skb1Δ*, *rad52-R45K*, *skb1Δ rad52-R45K*, *rad51Δ*, *skb1Δ rad51Δ*, *rad52-R45K rad51Δ*, and *skb1Δ rad52-R45K rad51Δ* strains (TNF5369, 5772, 6599, 8324, 5411, 5788, 7122, and 8345). The two-tailed Mann-Whitney test. Numerical data underlying **a**, **c**, and **e** are provided in Table A in Supplementary Data 1. Uncropped gel images are shown in Supplementary Fig. 5.

**Skb1 promotes Rad52-dependent isochromosome formation via its arginine methyltransferase activity.** Skb1 has been implicated in a wide range of pathways, including cell morphology and cell cycle regulation. Skb1 interacts with Slf1 and localizes to cell cortical nodes depending on Slf1, promoting rod-like cell morphology of fission yeast[40,41]. The DYRK-family kinase Pom1 negatively regulates cell cycle progression to ensure that cells grow to a certain size before entering mitosis[42]. Genetic evidence shows that Skb1 acts in the Pom1 pathway to regulate the cell cycle independently of its methyltransferase activity[43]. To ask whether Skb1 promotes isochromosome formation through these pathways, we disrupted *slf1* or *pom1* genes and determined the GCR rates of the mutant strains (Fig. 6a). Unlike *skb1Δ*, *slf1Δ* and *pom1Δ* slightly increased GCR rates in the wild-type background and neither reduced GCR rates in the *rad51Δ* background, showing that Skb1 promotes isochromosome formation through the function independent of Slf1 or Pom1.

Skb1 is the homolog of PRMT5 arginine methyltransferase (RMTase)[44,45]. The RMTase domain structure of *S. pombe* Skb1 predicted by AlphaFold methods[54,55] is very similar to the crystal structure of *C. elegans* PRMT5 RMTase domain (Fig. 6b) and to the predicted structures of *H. sapiens*, *A. thaliana*, and *S. cerevisiae* Skb1/PRMT5 homologs (Supplementary Fig. 3). The *skb1-A377V* mutation that we isolated is present in the RMTase domain (Figs. 1b, 6b), but the mutant phenotype was not as prominent as *skb1Δ* (Fig. 6c). To determine whether the RMTase activity is essential for Skb1 to promote isochromosome formation, we mutated the Skb1 residues F319, E422, and E431 equivalent to those important for the in vitro RMTase activity of *C. elegans* PRMT5[57]. We found that the *skb1-F319Y* and *skb1-E422A,E431A* mutations strongly reduced GCR rates in *rad51Δ* cells (Fig. 6c). Importantly, the same *skb1-E422A,E431A* mutation does not interfere with its role in cell cycle regulation[43]. PFGE analysis showed that, like *skb1Δ*, *skb1-F319Y* reduced isochromosomes but not chromosomal truncations in *rad51Δ* cells (Figs. 2c,

6d, and Table 1). We also examined the relationship between Skb1 and Rad52 and found that skb1Δ did not significantly reduce GCR rates in rad52-R45K or rad52-R45K rad51Δ cells (Fig. 6e). Together, these data indicate that Skb1 promotes Rad52-dependent isochromosome formation via its RMTase activity.

## Discussion

GCRs occur using repetitive sequences present in the eukaryote genome. However, the mechanism of repeat-mediated GCRs is largely unknown. Here, we found that the evolutionarily conserved Srr1 and Skb1 promote isochromosome formation using inverted DNA repeats at the centromere. Remarkably, srr1 increased DNA damage sensitivity in rad51Δ cells but did not abolish checkpoint response, suggesting that Srr1 promotes Rad51-independent DNA repair prone to GCRs. The srr1 and rad52 mutations additively, whereas skb1 and rad52 epistatically reduced GCR rates. Contrary to srr1 mutations, skb1Δ did not increase DNA damage sensitivity and reduced chromosome loss in rad51Δ cells. Through RMTase activity, Skb1 might form DNA structures leading to Rad52-dependent GCRs.

Srr1 and Skb1 promote isochromosome formation at the centromere. We found the srr1-W157R and skb1-A377V mutations in the same clone that exhibits reduced GCR levels (Fig. 1c). Either srr1 or skb1 deletion reduced GCR rates in rad51Δ cells (Fig. 1d), indicating that both Srr1 and Skb1 promote GCRs. Srr1 seems more integral to GCRs than Skb1 because srr1Δ but not skb1Δ reduced GCR rates even in the presence of Rad51 (Fig. 1d). Physical analysis of GCR products showed that Srr1 and Skb1 are responsible for ~90% of iso-chromosomes produced in rad51Δ cells but dispensable for chromosomal truncations (Fig. 2c). Given the fact that break-points of the isochromosomes are present in the centromere repeat[20], these results indicate that Srr1 and Skb1 promote repeat-mediated GCRs at the centromere. Whether Srr1 or Skb1 causes GCRs outside the centromere remains to be elucidated. The srr1 and skb1 mutations additively reduced GCR rates in rad51Δ cells (Fig. 1d, e), suggesting that Srr1 and Skb1 have non-overlapping roles in GCRs (see below).

How does Srr1 promote isochromosome formation? We propose that Srr1 promotes DNA damage repair prone to GCRs. In the wild type, Rad51 primally repairs detrimental DNA damage, such as DNA double-strand breaks, and safe-guards chromosome integrity. However, in the absence of Rad51, DNA damage left unrepaired will be channelled into other repair pathways, such as the Rad52-dependent recombi-nation pathway[29,30]. Srr1 is involved in Rad51-independent DNA damage response, as the srr1 and rad51 mutations addi-tively increased the sensitivity to MMS, HU, or CPT (Figs. 3a, 5e). Srr1 was not essential for MMS-induced cell cycle arrest and Chk1 phosphorylation (Fig. 3b, c), suggesting that Srr1 promotes DNA damage repair rather than checkpoint response. This is consistent with the fact that the checkpoint kinases Chk1 and Rad3 suppress but do not promote GCRs[20,58]. Like rad52Δ and rad52-R45K[32], srr1Δ eliminated ~90% of isochromosomes in rad51Δ cells (Fig. 2c), showing that Srr1 and Rad52 are essential for the major pathway of isochromosome formation. However, Srr1 and Rad52 may also have non-overlapping roles in GCRs, as srr1 and rad52 mutations additively reduced GCR rates (Fig. 4a). This is supported by the additive effect of srr1 and skb1 (Fig. 1d, e) and the epistatic effect of skb1 and rad52 on GCR rates (Fig. 6e). srr1Δ caused synthetic growth defects with rad52Δ and accumulated spontaneous foci of Rpa2 and Rad52 (Fig. 4b–e). srr1Δ increased the doubling time and pro-duced small colonies compared to the wild type (Table E in

Supplementary Data 1; Supplementary Fig. 2b). Srr1 may pro-mote the repair of spontaneous DNA damage. The hypersen-sitivity of srr1 cells to DNA-damaging agents supports this, but we do not exclude the possibility that Srr1 suppresses the for-mation of spontaneous DNA damage. Although its biochemical function remains unknown, we found the SRR1-like domain critical for GCRs and DNA damage repair. The srr1-D111A,P112A and srr1-H148A mutations altering evolutionarily conserved residues in the SRR1-like domain strongly reduced GCR rates and increased sensitivity to the DNA-damaging agents, which interfere with the fork progression and eventually create DNA breaks (Fig. 5). The N-terminal extension of Srr1 is predicted intrinsically disordered in ref. [56] and contains posi-tively charged residues in fission yeast and other organisms including humans, suggesting its role in the interaction with other molecules. Interestingly, protein structure comparison using Dali server[59] suggests that the SRR1-like domain has structural similarity to a set of proteins, including methyl-transferases (Supplementary Data 2). The knockout of the Srr1 homolog SRRD in mouse cells impairs DNA replication[37], suggesting that mammalian Srr1 also plays a role in the repair of spontaneous DNA damage created during DNA replication. In plants and mammals, SRR1/SRRD affects the transcription of the genes involved in the circadian rhythm[37,38,60], raising the possibility that Srr1 promotes GCRs and DNA damage response through transcriptional regulation. Interestingly, the srr1/ber1 mutation in budding yeast causes synthetic growth defects with mutations in the centromere proteins, including the centromere-specific histone H3 variant Cse4/CENP-A[39]. Although, unlike budding yeast srr1Δ/ber1Δ cells, fission yeast srr1Δ cells were not hyper-resistant to a microtubule-destabilizing drug, it is still possible that Srr1 promotes iso-chromosome formation by affecting the centromere structure. Srr1 has been localized to both the nucleus and cytosol in fis-sion yeast[61], but whether Srr1 localizes to the centromere remains unknown. Future works are required to address how Srr1 promotes isochromosome formation and DNA damage response.

It has been shown that a mutation in the Fbh1 helicase sup-presses growth and recombination defects of rad52Δ cells[62], raising the possibility that spontaneous fbh1 mutations have been introduced into our rad52Δ strain and affect GCRs. However, we previously showed no fbh1 mutations in our rad52Δ rad51Δ strain[36] used to determine GCR rates. We also confirmed no fbh1 mutations in all the rad52 mutant strains and the srr1Δ rad51Δ strain used in this study (Supplementary Data 3). Furthermore, the fbh1 deletion did not significantly affect isochromosome formation in rad51Δ cells (Supplementary Fig. 4).

Skb1 has been implicated in many pathways. However, our data show that Skb1 promotes isochromosome formation independent of its role in cell morphology and cell cycle reg-ulation mediated by Slf1 and Pom1, respectively (Fig. 6a). Skb1 and Slf1 bind each other and are mutually required for their localization to cell cortical nodes[40]. The cortical node localiza-tion is not essential for Skb1 to promote isochromosome for-mation, as slf1Δ did not reduce GCR rates. In rad51Δ cells, rad52-R45K reduces isochromosome formation and increases DNA damage sensitivity[32]. The pcn1-K107R mutation in a DNA sliding clamp PCNA and rad52-R45K epistatically reduce iso-chromosome formation, but pcn1-K107R does not increase DNA damage sensitivity[36]. Like pcn1-K107R, skb1Δ impairs Rad52-dependent isochromosome formation (Fig. 6e) but does not increase DNA damage sensitivity (Fig. 3a). In concert with PCNA, Skb1 might form DNA structures leading to Rad52-dependent GCRs. rad51Δ cells showed elevated levels of chro-mosome loss and GCRs, probably due to the inability to repair

spontaneous DNA damage. As in the case of GCRs (Fig. 1d), *skb1Δ* reduces chromosome loss in *rad51Δ* but not in wild-type cells (Fig. 3e), showing that Skb1 causes GCRs and chromosome loss in *rad51Δ* cells. Skb1 is the fission yeast homolog of PRMT5 arginine methyltransferase (RMTase)[44,63]. PRMT5 is overexpressed in many malignant tumours, including breast cancer, and plays a role in the development of cancer[64]. Interestingly, the proteins affecting chromatin structure and replication, such as histones, Fen1 DNA-flap endonuclease, and p53, have been found as PRMT5 substrates[65–68]. Skb1 also interacts with Shk1, a p21-activated kinase (PAK) that nega- tively regulates cell cycle progression[69]. Mutating the residues essential for the RMTase activity:[57] *skb1-F319Y* and *skb1-E422A,E431A* strongly reduced GCR rates, suggesting that Skb1 promotes GCRs through the RMTase activity. It is crucial to identify the key substrate(s) of Skb1 RMTase that induces chromosome instability in the future. Nonetheless, our findings paved a new avenue for elucidating the mechanism of GCRs at the centromere.

## Methods

**Genetic procedures and yeast strains.** The fission yeast strains used in this study are listed in Supplementary Table 1 and are available from the corresponding author upon request. DNA primers used in this study are listed in Supplementary Table 2. Cells were grown in yeast extract (YE) medium or Edinburgh minimal medium 2 (EMM)[70] at 30 °C unless otherwise indicated. Amino acids or bases were added at a final concentration of 225 mg L$^{-1}$. Yeast nitrogen base (YNB) medium contained 7 g L$^{-1}$ of yeast nitrogen base (BD Difco, BD 291940) and 20 g L$^{-1}$ glucose. YNB medium is supplemented with 1 g L$^{-1}$ 5-fluoroorotic acid (Apollo Scientific, PC4054) and 56 mg L$^{-1}$ uracil (Nacalai Tesque, 35824–82) to prepare 5-FOA media. Solid media contained 1.5% agarose (Nacalai Tesque, 01028–85). Yeast transformation was performed by the lithium acetate/PEG method[71]. Yeast cells were grown in YE or YE3S medium until log phase ($1–2 \times 10^7$ cells mL$^{-1}$) and harvested by centrifugation. Cells were washed once with sterilized water and twice with 1 mL LiAc/TE buffer (0.1 M lithium acetate, 10 mM Tris-HCl (pH 7.5), 1 mM EDTA). Cells were suspended in LiAc/TE buffer at $> 2 \times 10^9$ cells mL$^{-1}$. 100 μL of the cell suspension was mixed with 5 μL of salmon sperm DNA (10 mg mL$^{-1}$) and the introducing DNA and incubated at room temperature for 10 min. After adding 260 μL of PEG/LiAc/TE (40% PEG4000, 0.1 M lithium acetate, 10 mM Tris-HCl (pH 7.5), 1 mM EDTA), the tube was further incubated for 30 min with rotation. After adding 43 μL of dimethyl sulfoxide (DMSO), the tube was incubated at 42 °C for 5 min. Cells were harvested by centrifugation at $503 \times g$ for 30 s, suspended in YE or YE3S media, and plated on non-selective media. After one day of incubation, the cells were replica plated onto a medium supplemented with G418 (Nacalai Tesque, 09380–86) or hygromycin B (Nacalai Tesque, 09287–87) at a final con- centration of 100 μg mL$^{-1}$ or clonNAT (Werner BioAgents, 5.001.000) at 50 μg mL$^{-1}$ to select the transformants. We did not pick up exceptionally large colonies of *rad52* mutants because they can contain an *fbh1* mutation[62].

To search for the genes causing GCRs in *rad51Δ* cells, we introduced random mutations into yeast essentially, as described previously in ref. [32]. Nitrous acid was used as the mutagen because it efficiently introduces mutations in DNA- repair deficient cells, as in wild-type cells[72]. *rad51Δ* cells containing ChL$^C$ (TNF5411) grown in EMM were collected at the log phase ($5 \times 10^6$ cells mL$^{-1}$), suspended in water, and kept overnight at 4 °C. After centrifugation, cells were suspended in 0.8 mL of 0.01 M nitrous acid solution prepared before use by dissolving sodium nitrate (Wako, 195–20562) in 0.5 M sodium acetate (pH 4.8) and incubated at room temperature for 20 min. After adding an equal volume of the stop buffer (3.6% Na$_2$HPO$_4$·12H$_2$O and 1% yeast extract) to the cell suspension, cells were plated on EMM+UA plates. The plating efficiency of the mutagenized cells determined using EMM plates was around 10%. 24,000 independent clones were incubated as patches on EMM+UA plates for 2–3 d at 30 °C and then transferred onto 5-FOA+UA plates to semi-quantitatively determine the rate of uracil auxotroph, resulting from GCR or a point mutation in the *ura4* gene. 80 clones produced reduced numbers of colonies on 5-FOA +UA plates. PFGE analysis showed that six contained the aberrant sizes of the parental ChL$^C$. The remaining 74 clones were crossed with wild-type cells containing ChL$^C$. Three clones reproducibly exhibited reduced GCR rates. Deep sequencing of genomic DNA was carried out using MiSeq (Illumina, San Diego, CA), and the mutations were identified by pooled-linkage analysis[73,74]. Nucleotide sequence data of the parental strain and a pool of nine mutant segregants obtained by backcrossing one of the three clones are available in the DDBJ Sequenced Read Archive under the accession numbers DRX042095 and DRX042098, respectively.

The *srr1::kanMX6* strain was created by two rounds of polymerase chain reaction, PCR[75]. srr1-kan3 and srr1-kan5 primers were designed such that the 3' side was complementary to *srr1* and the 5' side was complementary to the *kanMX6*

gene on pFA6a-kanMX6 plasmid[76]. In the first round of PCR, the *srr1* flanking regions of 0.5 kb were amplified, using srr1-kan3/srr1-3 or srr1-kan5/srr1-1 primer pairs and fission yeast genomic DNA as a template. The second round of PCR was carried out in the presence of the two PCR fragments, pFA6a-kanMX6, and srr1-1/ srr1-3 primers, to produce the DNA fragment containing the *srr1::kanMX6* construct. The 2.5 kb PCR fragment was introduced into yeast.

The *skb1::kanMX6* (or *skb1::hphMX6*) strain was created similarly as described above. In the first round of PCR, *skb1* flanking regions of 0.5 kb each were amplified, using skb1-1/skb1-kan5 or skb1-kan3/skb1-2 primer pairs. The second round of PCR was carried out in the presence of the two PCR fragments, pFA6a- kanMX6 (or pFA6a-hphMX6), and skb1-1/skb1-2 primers, to produce the DNA fragment containing the *skb1::kanMX6* or *skb1::hphMX6* construct. The 2.5 or 2.7 kb PCR fragment was introduced into yeast.

To create the *srr1-W157R* mutant strain, the *ura4$^+$* gene was introduced into the *srr1* gene in *ura4-D18* cells, making *ura4$^+$:srr1* cells. In the first round of PCR, 0.5 kb regions were amplified using srr1-F1/srr1-ura4AN5 and srr1-ura4AN3/srr1- R1 primer pairs. The second round of PCR was carried out in the presence of the two PCR fragments, a plasmid containing the *ura4$^+$* genomic fragment, and srr1- F1/srr1-R1 primers. The 3 kb PCR fragment was transformed into yeast cells, and the *ura4$^+$* transformants were selected on EMM media. Next, the genomic region containing the *srr1-W157R* mutation was amplified by PCR using srr1-1/srr1-R1 primers and the template DNA prepared from the *srr1-W157R skb1-A377V* mutant isolated in the screening. The 1.5 kb PCR product was introduced into the *ura4$^+$:srr1* strain, and *ura4$^-$* transformants were selected on 5-FOA plates. DNA sequencing confirmed the integration of the *srr1-W157R* mutation and no additional mutations. Following the construction of the *ura4$^+$:skb1* strain, the *skb1-A377V* strain was created in the same way, except skb1-F1/skb1-2 primers were used.

The *srr1-H148A* mutant fragment was also created in two rounds of PCR. First, 0.6 and 0.5 kb fragments were amplified using srr1-H148A-F/srr1-R1 and srr1- H148A-R/srr1-F1 primer pairs, respectively, and fission yeast genomic DNA as a template. Both the srr1-H148A-F and srr1-H148A-R primers contain the *srr1- H148A* mutation. The two overlapping PCR fragments were joined in the second round of PCR in the presence of srr1-F1/ srr1-R1 primers. The 1.1 kb product was introduced into the *ura4$^+$:srr1* strain, and *ura4$^-$* transformants were selected on 5-FOA plates. The *srr1-D111A,P112A* mutant was created in a similar manner. In the first round of PCR, 0.7 and 0.5 kb fragments were amplified using srr1-DPAA- F/srr1-R1 and srr1-DPAA-R/srr1-F1 primer pairs, respectively. The two PCR fragments were joined in the second PCR reaction containing srr1-F1/srr1-R1 primers. The 1.1 kb product was introduced into the *ura4$^+$:srr1* strain. *skb1-F319Y* and *skb1-E422A,E431A* mutants were generated in the same way using skb1-F1/ skb1-F319Y-R and skb1-F319Y-F/skb1-R1 primer pairs, and skb1-F1/skb1- doubleE-R and skb1-doubleE-F/skb1-R1 primer pairs, respectively. Correct integration of the point mutations, but no additional mutations were confirmed by Sanger sequencing.

**GCR rates.** GCR rates were determined by the fluctuation test as described pre- viously in ref. [36]. Yeast cells were incubated on EMM+UA plates for 6–8 days. Single colonies were used to inoculate 10 mL EMM+UA liquid medium. After 1–2 days of incubation, cells were plated onto 5-FOA+UA and YNB+UA plates. After 5–9 days of incubation, colonies formed on 5-FOA+UA and YNB+UA were counted to determine the number of Leu$^+$ Ura$^-$ and Leu$^+$ cells, respectively. About six colonies from each 5-FOA+UA plate were streaked onto EMM+UA to examine the colony size and then transferred to EMM+U plates to inspect adenine auxotrophy. The number of Leu$^+$ Ura$^-$ Ade$^-$ cells, indicative of GCR, was obtained by subtracting the number of Leu$^+$ Ura$^-$ Ade$^+$ cells from that of Leu$^+$ Ura$^-$ cells. The GCR rates per cell division were determined as described in ref. [77]. When we started yeast cultures, we randomly picked up colonies of different sizes. We recovered both large and small colonies on 5-FOA+UA plates for PFGE analyses, according to the ratio of their appearance.

**Chromosome loss rates.** Chromosome loss rates were determined by the fluc- tuation test as described previously in ref. [27]. A single colony formed on YE3S plates after 3–4 days incubation was suspended in sterilized water, and the cells were plated on YE plates. After 4–6 days incubation, white and red colonies were counted. The red colonies, indicative of *ade6* loss, were transferred to YE3S and EMM+UA plates to inspect leucine auxotrophy. Colonies grown on YE3S plates were replica plated on EMM+UL and EMM+AL plates to test adenine and uracil auxotrophy, respectively. The number of Leu$^-$ Ura$^-$ Ade$^-$ cells, indicative of ChL$^C$ chromosome loss, and the total number of colony-forming cells were used to obtain the chromosome loss rate per cell division[77].

**PFGE analysis of GCR products.** Each GCR clone was obtained from an inde- pendent culture to avoid multiple clones from the same parent. Cells were grown in YE3S medium at 25 °C for 12–24 h. $1 \times 10^8$ cells were harvested, suspended in 2.5 mL ice-cold 50 mM EDTA, and stored at 4 °C. The cells were centrifuged and resuspended in 1 mL CSE buffer (20 mM citrate phosphate, 1 M sorbitol, 50 mM EDTA (pH 5.6)). To prepare spheroplasts, 5 μL Zymolyase 20 T (Seikagaku, Tokyo, Japan, 25 mg mL$^{-1}$) and 5 μL lyzing enzyme (Sigma, St. Louis, Missouri,

25 mg mL$^{-1}$) were added to the cell suspension and incubated at 30 °C for 20 to 50 min. Spheroplasts were harvested by centrifugation at $33 \times g$ for 10 min at 4 °C, and the pellet was suspended in 140 μL CSE buffer. An equal volume of 1.6% low melting agarose gel (Nacalai Tesque, 01161–12) pre-heated at 50 °C was added to the cell suspension and distributed into molds. The agarose plugs were incubated at 4 °C for 20 min. The plugs were incubated in SDS-EDTA solution (1% SDS, 0.25 M EDTA) at 60 °C for 2 h and then in ESP solution (0.5 M EDTA, 1% N-lauroyl sarcosine, 1.5 mM calcium acetate) supplemented with 0.5 mg mL$^{-1}$ proteinase K (Nacalai Tesque, 39450–01–6) at 50 °C overnight. The plugs were transferred into another ESP solution supplemented with 0.5 mg mL$^{-1}$ proteinase K and incubated at 50 °C for an additional 8 h. The plugs were stored in TE buffer (10 mM Tris-HCl (pH 8.0), 1 mM EDTA) at 4 °C. Chromosomal DNAs were resolved using a CHEF-DRII pulsed-field electrophoresis system (Bio-Rad, Hercules, California). PFGE ran at 4.2 V cm$^{-1}$ with a pulse time of 40 to 70 s for 24 h, at 4 °C in 0.5× TBE buffer (89 mM Tris-borate, 2 mM EDTA) using 0.55% Certified Megabase agarose gel. DNA was stained with 0.2 μg mL$^{-1}$ ethidium bromide (EtBr) (Nacalai Tesque, 14631–94) and detected using a Typhoon FLA9000 gel imaging scanner (GE Healthcare, Chicago, Illinois) or GelDoc Go imaging system (Bio-Rad, Hercules, CA). Gel images were processed using ImageJ2 2.9.0 (NIH, United States) or Adobe Photoshop Elements 2020 (Adobe, San Jose, CA).

**Serial dilution assay.** A single colony formed on YE (or YE3S) plates after 3–4 days incubation was used to inoculate 2 mL YE (or YE3S) liquid media. The 2 mL overnight culture was used to prepare 10 mL log-phase cultures. Five-fold serial dilutions of the indicated strains were prepared with sterilized water. 6 μL from each dilution was spotted onto YE (or YE3S) plates supplemented with the indicated concentration of MMS, HU, CPT, or TBZ. The plates were incubated for 3−5 days at 30 °C. Images were taken using GT-X800 (Epson, Nagano, Japan) and processed using Adobe Photoshop Elements 2020 (Adobe, San Jose, CA).

**Western blot.** Cell extracts were prepared using an alkaline lysis method[78]. $1 \times 10^8$ cells from log-phase YE cultures were collected, washed with 1 ml H$_2$O, and suspended in 300 μl H$_2$O. After adding 300 μl of 0.6 M NaOH, the cell suspension was incubated at 30 °C for 5 min with the tube rotating. After centrifugation at 6000 rpm for 3 min (TOMY, MX-201), the alkali-treated cells were suspended with 140 μ of SDS sample buffer (60 mM Tris-HCl (pH6.8), 5% glycerol, 4% sodium dodecyl sulfate, 4% β-mercaptoethanol, 0.005% bromophenol blue) and incubated at 95 °C for 3 min. Cell extracts were recovered from the supernatant after centrifugation at 15,000 rpm for 1 min (TOMY, Kitman), separated by 8% sodium dodecyl sulfate-polyacrylamide gel electrophoresis (SDS-PAGE) (acrylamide to bisacrylamide ratio, 37.5:1), and transferred onto PolyScreen PVDF Transfer Membrane (Perkin Elmer, NEF1002001PK). To detect Chk1-HA, a mouse monoclonal antibody against the HA tag (16B12, Abcam, Cambridge, MA) (1:2000) and peroxidase AffiniPure goat anti-mouse IgG (heavy+light) (Jackson ImmunoResearch Laboratories, 115−035−146) (1:10,000) were used as the primary and secondary antibodies, respectively. The blots were developed using Supersignal West Femto substrate (ThermoScientific, 34095). Images were captured using ImageQuant LAS 500 (GE Healthcare).

**Septation index.** MMS was added to log-phase EMM cultures to a final concentration of 0.01%. At 0, 2, 4, 6, and 8 h after adding MMS, cells were harvested, suspended in 70% ethanol, and stored at 4 °C. Cells were suspended in PEMS buffer (100 mM PIPES (pH6.9), 1 mM EGTA, 1 mM MgSO4, 1 M sorbitol), containing 2 μg mL$^{-1}$ 4′,6-diamidino-2-phenylindole (DAPI) and 4 μg mL$^{-1}$ calcofluor. The cell suspension was mixed with a mounting medium (90% glycerol, 1 mg mL$^{-1}$ n-propyl gallate, 1 mg mL$^{-1}$ 1,4-phenylenediamine dihydrochloride, 0.1× phosphate buffered saline) on poly-L-lysine coated coverslip. Using a fluorescence microscope (BX51, Olympus, Tokyo, Japan) with a 100× objective (NA = 1.40, Olympus, Tokyo, Japan), nuclei and septa were visualized with DAPI and calcofluor, respectively. Images were obtained using a charge-coupled device camera (DP72, Olympus, Tokyo, Japan) and processed using cellSens Standard 2.3 (Olympus) and Adobe Photoshop Elements 2020 (Adobe, San Jose, CA).

**Fluorescent microscopy detecting Rpa2-mCherry and Rad52-GFP foci.** Exponentially growing cells in EMM (Rpa2-mCherry) or YE (Rad52-GFP) medium were collected, seeded on glass-bottom dishes (Matsunami Glass, Osaka, Japan, D11130H), and observed using a DeltaVision Personal fluorescence microscopy system (GE Healthcare), which is based on an Olympus wide-field IX71 fluorescence microscope equipped with a CoolSNAP HQ2 CCD camera (Photometrics, Tucson, Arizona) and an oil-immersion objective lens (UAPO 40×; NA = 1.35; Olympus, Tokyo, Japan). The percentage of nuclei with Rpa2-mCherry and Rad52-GFP foci was obtained by counting the nuclei containing at least one Rpa2-mCherry and Rad52-GFP focus, respectively, using ImageJ2 2.9.0 (NIH, United States). >300 nuclei were counted in each experiment. Three independent experimental values and their means were shown in the graph using GraphPad Prism 9 for MacOS (GraphPad Software, San Diego, CA). Images were processed using ImageJ2 2.9.0 or Adobe Photoshop Elements 2020.

**Rad52-AID2 depletion assay.** Exponentially growing cells of the indicated strains were prepared and five-fold serially diluted with sterilized water. 6 μL from each dilution was spotted onto YE plates supplemented with DMSO or 200 nM 5′a-IAA (Tokyo Chemical Industry, A3390)[51]. The plates were incubated for 2 days. Images were taken using GT-X800 (Epson, Nagano, Japan) and processed using Adobe Photoshop Elements 2020 (Adobe, San Jose, CA).

**Statistics and reproducibility.** The two-tailed Mann-Whitney and the two-tailed Fischer's exact tests were performed using GraphPad Prism 9 for MacOS. The two-tailed student's $t$-test and Pearson's Chi-square test were performed using Microsoft Excel for Mac 16.72. The sample size used to derive each statistic was provided in the figure legend or in the supplementary information.

**Reporting summary.** Further information on research design is available in the Nature Portfolio Reporting Summary linked to this article.

## Data availability

The data supporting the findings of this study are included in the paper, Supplementary Information (Supplementary Figs. 1–6 and Supplementary Tables 1-2), and Supplementary Data 1-3. DNA sequence data of the parental strain and a pool of the mutant segregants are available in the DDBJ Sequenced Read Archive under the accession numbers DRX042095 and DRX042098, respectively, at the following URL: https://ddbj.nig.ac.jp/resource/bioproject/PRJDB4206. All raw datasets are available from the corresponding author upon reasonable request.

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

## Acknowledgements
We thank Akiko Okita, Jie Su, and Yukiko Kubota for their critical comments on the manuscript and Hirofumi Ohmori and Keiko Kayahara for their technical assistance. We also thank Adam T. Watson and Antony M. Carr for sharing the AID2 system. This work was supported by JSPS KAKENHI Grant Numbers 221S0002, JP23570212, JP26114711, 18K06060, 21H02402, and the Uehara Memorial Foundation Grant Number 202120462 to TN.

## Author contributions
P.M., N.T., and T.N. conceived the study. P.M., N.T., and Z.P. performed most experiments with technical help from R.X., Y.K., K.O., and T.N. Deep sequencing was performed by H.T., Y.O., and T.H. The manuscript was written by T.N. and P.M. and approved by all the authors.

## Competing interests
The authors declare no competing interests.
