## [Peer Review File · Communications Biology]

Reviewers' comments:

Reviewer #1 (Remarks to the Author):

This manuscript by Mongia et al. report on intriguing experiments aimed at understanding the genesis and regulation of isochromosome formation (via Gross Chromosomal-Rearrangements; GRC within repetitive centromeres) in fission yeast.

Via a genetic screening approach the authors identify to factors, Srr1 and Skb1, whose loss reduces the formation of GCR in rad51del backgrounds.

The follow-up experiments convincingly show that these two factors contribute the prevention of GRCs. Although many aspects of the function of these genes/functions are still to be uncovered, this manuscript is of high quality, reports on interesting findings using well-designed experiments, and is well-written. This reviewer thus strongly supports publication, after addressing the (minor) points listed below.

Minor points:

- 1) line 122. ' ..showing that srr1 and skb1 have non-overlapping..' Are these differences really 'showing' this at this point? "Support/hint" would be better? Maybe scale down this claim?
- 2) line 127/ figure 1e. These experiments show that the skb1-AV allele doesn't affect GCRs. If so, how do the authors explain that this allele was uncovered in (the highly selective) screen? What is different here that might explain these differences?
- 3) line 206/figure 4d: '..., suggesting a role for Srr1 in the rad52-independent pathway.'. Can this really be concluded here? Isn't the data merely suggesting that Rad52 foci are increased in Srr1del cells?
- 4) line 214, can the authors suggest/investigate (using AF2) what the SRR1-like domain might look like/what its function might be?
- 5) line 232: " ..via the RMTase activity." change to " ..via its RMTase activity."?
- 6) Figure 6: can the authors add some more AF2 models, also including other species (can be in Supp Figures)? Would be nice to see how similar is throughout evolution.
- 7) Figure 6c: Can the authors as a quantification of these data, as shown in figure 2c for example?
- 8) line 152: is the colon between 'isochromosome formation' and 'repeat-mediated GCRs' needed here? This sentence is confusing, please rephrase.
- 9) General question (which might be tested, or discussed in discussion): are Skb1 and/or Srr1 found at centromeres, as might be expected? Any evidence in other species?

Reviewer #2 (Remarks to the Author):

This paper continues the ongoing and fruitful project in the Nakagawa lab to understand the determinants of chromosome stability in the fission yeast centromere repeats. Previous work from this group has shown that recombination in the inner repeats is opposed by Rad51 protein so that rad51Δ mutants have increased GCR. In this study, they perform a genetic screen to isolate mutations that rescue the enhanced GCR phenotype of rad51Δ, and identify two conserved genes, srr1 and skb1. Loss of function alleles of these genes suppress the formation of isochromosomes in rad51Δ. While their identification is interesting in principle, mechanistic information here is very limited, and the study therefore seems somewhat incomplete and phenomenological for this audience.

Minor point: while "representative data" for their screen is presented Fig 1b, it is not clear what the phenotype looks like in the mutants under these replica conditions. Are any of those in 1b the mutants presented here?

Epistasis analysis suggests that *srr1Δ* causes sensitivity to genotoxins by itself, a phenotype that is enhanced in the double mutant with *rad51Δ*. However, the authors do not provide any data as to the phenotype of the cells: for example, elongated and arrested would suggest a repair defect and chronic checkpoint activation, while shorter cells would suggest a checkpoint bypass.

Of note, *srr1Δ* is synthetically lethal with *rad52Δ*.

However, while conserved, the gene is relatively uncharacterized in any system, and this study gives us only limited new insights into its function. While the authors speculate that it may play a role through transcriptional regulation or through modifying DNA damage response during DNA replication, no further epistasis molecular or genetic analysis is provided. Evidence that *srr1Δ* mutants have increased Rad52 foci is not particularly significant, and it would be of interest to also examine RPA foci.

Minor point: *rad52Δ* mutants are notorious for picking up suppressors in *fbh1Δ*. Can the authors be sure that they aren't dealing with *fbh1Δ* mutations in their *rad52Δ* strains, and have they examined the role of *fbh1Δ* in their system/with *srr1*?

The epistasis with *skb1*, an arginine methyltransferase, suggests it operates independently of the *srr1Δ* pathway. It does not show intrinsic genotoxin instability by itself or enhance the phenotype of *rad51* or *rad52* mutants so it's not clear how direct this effect may be. Genetic and physical studies elsewhere implicate it with a wide range of pathways including cell shape, cell cycle, and chromatin structure. Again, we have no real insights from this work as to the mechanism or specificity of Skb1 function in this pathway.

Reviewer #3 (Remarks to the Author):

Mongia et al : Fission yeast Srr1 and Skb1 promote isochromosome formation at the centromere.

Here the authors have sought factors that cause GCRs in a *rad51Δ* mutant strain and find Srr1 (Ber1) and Skb1 (the orthologue of PRMT5). Using a previously described minichromosome ChLC they show that these factors promote isochromosome formation at centromeres and that mutations in these factors, including mutations in the arginine methyltransferase domain of SKb1 lead to elevated levels of mutations in a *rad51Δ* background. Further analysis reveals Srr1 to be required for the DNA damage response and to promote DNA damage independently of Rad51, and to have both overlapping and non-overlapping roles with Rad52 in GCR formation. Further the SRR1-like domain is required for this function in SRR1, while a beautiful structural analysis suggests that the Skb1 arginine methyltransferase acts in the Rad52-dependent GCR pathway.

The manuscript is very well and clearly written, and the data they present are original, very clear and largely support the conclusions drawn. However, the data, while interesting, remain somewhat preliminary and raise more questions than they really answer. For example, there are no mechanistic insights into how these two mutations limit GCRs in the various backgrounds; whether they limit GCRs through impacting the initiation of these spontaneous GCR-inducing events, or whether they are required in the processing mechanisms leading to GCR formation? Further, while these data suggest that Skb1 functions in the Rad52 GCR pathway, it is perhaps surprising that loss of Skb1 does not show any increased sensitivity to any of the DNA damaging agents tested, which *rad52Δ* clearly does. Lastly, it is not entirely clear why both of these mutations are included together in the same manuscript, as clearly the double mutant has increased sensitivity suggesting non-overlapping functions leading to GCR's. In summary, the story is novel and encouraging but would benefit from further mechanistic insights.

Point-by-point Response to Reviewers' comments:

Thank you for your time considering our manuscript. I appreciate insightful comments and constructive suggestions from the reviewers. Please find the point-by-point response as follows. The parts we added or changed according to the reviewers' comments are highlighted in blue in the text.

Reviewer #1 (Remarks to the Author):

This manuscript by Mongia et al. report on intriguing experiments aimed at understanding the genesis and regulation of isochromosome formation (via Gross Chromosomal-Rearrangements; GRC within repetitive centromeres) in fission yeast.

Via a genetic screening approach the authors identify to factors, *Srr1* and *Skb1*, whose loss reduces the formation of GCR in *rad51del* backgrounds.

The follow-up experiments convincingly show that these two factors contribute the prevention of GRCs. Although many aspects of the function of these genes/functions are still to be uncovered, this manuscript is of high quality, reports on interesting findings using well-designed experiments, and is well-written. This reviewer thus strongly supports publication, after addressing the (minor) points listed below.

Minor points:

1) line 122. ' ..showing that *srr1* and *skb1* have non-overlapping..' Are these differences really 'showing' this at this point? "Support/hint" would be better? Maybe scale down this claim?

Line 129. Yes, we changed "showing" to "suggesting".

2) line 127/ figure 1e. These experiments show that the *skb1-AV* allele doesn't affect GCRs. If so, how do the authors explain that this allele was uncovered in (the highly selective) screen? What is different here that might explain these differences?

As the reviewer suggested, we explained why our genetic screening uncovered the *skb1-A377V* allele.

Line 134-138. "Unlike *skb1Δ*, *skb1-A377V* ... both mutations in the same clone."

3) line 206/figure 4d: '...', suggesting a role for *Srr1* in the *rad52*-independent pathway.". Can this really be concluded here? Isn't the data merely suggesting that *Rad52* foci are increased in *Srr1del* cells?

Yes, we revised the conclusion.

Line 237-238. " showing that *Srr1* suppresses the *Rad52* focus formation"

4) line 214, can the authors suggest/investigate (using AF2) what the *SRR1*-like domain might look like/what its function might be?

Thank you for the suggestion, we added new panels Fig. 5b and c showing the predicted structure of the *Srr1* protein and discussed the molecular function.

Line 241-244. "The *Srr1* structure predicted ... N- and C-termini (Fig. 5b and c)."

Line 326-331. "The N-terminal extension ... (Table F in Supplementary Data 1)."

5) line 232: " ..via the RMTase activity." change to " ..via its RMTase activity."?

Line 279. Yes, we revised it.

6) Figure 6: can the authors add some more AF2 models, also including other species (can be in Supp Figures)? Would be nice to see how similar is throughout evolution.

Yes, we created a new Supplementary Fig. 3, showing AlphaFold models of the Skb1 homologs in *S. pombe*, *C. elegans* (the crystal structure), *H. sapiences*, *A. thaliana*, and *S. cerevisiae*.

Line 266-267. “and to the predicted structures ... (Supplementary Fig. 3).”

7) Figure 6c: Can the authors as a quantification of these data, as shown in figure 2c for example?

Yes, the quantification is shown in Fig. 2c.

8) line 152: is the colon between 'isochromosome formation' and 'repeat-mediated GCRs' needed here? This sentence is confusing, please rephrase.

Yes, we removed the colon and rephrased the sentence.

Line 165, “isochromosome formation mediated by centromere repeats”.

9) General question (which might be tested, or discussed in discussion): are Skb1 and/or Srr1 found at centromeres, as might be expected? Any evidence in other species?

Yes, we discussed the Srr1 and Skb1 localization, respectively.

Line 340-341. “Srr1 has been localized to ... the centromere or not.”

Line 353-356. “Skb1 and Sif1 bind ... reduce GCR rates.”

Reviewer #2 (Remarks to the Author):

This paper continues the ongoing and fruitful project in the Nakagawa lab to understand the determinants of chromosome stability in the fission yeast centromere repeats. Previous work from this group has shown that recombination in the inner repeats is opposed by Rad51 protein so that *rad51Δ* mutants have increased GCR. In this study, they perform a genetic screen to isolate mutations that rescue the enhanced GCR phenotype of *rad51Δ*, and identify two conserved genes, *srr1* and *skb1*. Loss of function alleles of these genes suppress the formation of isochromosomes in *rad51Δ*. While their identification is interesting in principle, mechanistic information here is very limited, and the study therefore seems somewhat incomplete and phenomenological for this audience.

Minor point: while “representative data” for their screen is presented Fig 1b, it is not clear what the phenotype looks like in the mutants under these replica conditions. Are any of those in 1b the mutants presented here?

We performed the replica plating assay using the parental *rad51Δ* and the *rad51Δ* clone containing *srr1-W157R* and *skb1-A377V* mutations (Fig. 1c), showing the effect of the *srr1 skb1* double mutation under these replica conditions.

Line 121-123. “The replica plating assay ... the 5-FOA+UA plate (Fig. 1c).”

Epistasis analysis suggests that *srr1Δ* causes sensitivity to genotoxins by itself, a phenotype that is enhanced in the double mutant with *rad51Δ*. However, the authors do not provide any data as to the phenotype of the cells: for example, elongated and arrested would suggest a repair defect and chronic checkpoint activation, while shorter cells would suggest a checkpoint bypass.

Thank you for the constructive suggestion. We performed additional experiments to see the effect of Srr1 on DNA damage checkpoint response (Fig. 3b and c) and found that Srr1 is not essential for cell cycle arrest (Fig. 3b) and Chk1 phosphorylation induced by MMS treatment (Fig. 3c). These new data show that Srr1 is not essential for DNA damage checkpoint activation.

Line 176-190. "Cell cycle arrest caused ... the checkpoint activation."

Line 309-312. "Srr1 was not essential ... do not promote GCRs."

Of note, *srr1* Δ is synthetically lethal with *rad52* Δ .

However, while conserved, the gene is relatively uncharacterized in any system, and this study gives us only limited new insights into its function. While the authors speculate that it may play a role through transcriptional regulation or through modifying DNA damage response during DNA replication, no further epistasis molecular or genetic analysis is provided. Evidence that *srr1* Δ mutants have increased Rad52 foci is not particularly significant, and it would be of interest to also examine RPA foci.

Thank you again for the suggestion. We performed additional experiments to detect Rpa2-mCherry foci in wild-type and *srr1* Δ cells (Fig. 4d) and found that *srr1* Δ increased the spontaneous formation of the Rpa2 foci. Given the increased damage sensitivity of *srr1* Δ cells, these data suggest a role of Srr1 in the repair of spontaneous DNA damage.

Line 229-236. "Given the hypersensitivity of *srr1* Δ ... single-stranded DNA in *srr1* Δ cells."

Line 317-321. "*srr1* Δ caused synthetic growth ... spontaneous DNA damage."

Minor point: *rad52* Δ mutants are notorious for picking up suppressors in *fbh1* Δ . Can the authors be sure that they aren't dealing with *fbh1* Δ mutations in their *rad52* Δ strains, and have they examined the role of *fbh1* Δ in their system/with *srr1*?

Previously, we have shown that there were no *fbh1* mutations in our *rad52* Δ *rad51* Δ strains used to determine GCR rates (Su *et al.*, *PLoS Genet.*, 2021, DOI:

10.1371/journal.pgen.1009671). We also performed DNA sequencing and found no *fbh1* mutations in the *srr1* Δ *rad51* Δ strain (Supplementary Data 2). Furthermore, we found that *fbh1* Δ does not affect isochromosome formation in the *rad51* Δ strain (Supplementary Fig. 2).

Line 344-350. "It has been shown that a ...formation in *rad51* Δ cells (Supplementary Fig. 2)."

The epistasis with *skb1*, an arginine methyltransferase, suggests it operates independently of the *srr1* Δ pathway. It does not show intrinsic genotoxin instability by itself or enhance the phenotype of *rad51* or *rad52* mutants so it's not clear how direct this effect may be. Genetic and physical studies elsewhere implicate it with a wide range of pathways including cell shape, cell cycle, and chromatin structure. Again, we have no real insights from this work as to the mechanism or specificity of Skb1 function in this pathway.

Skb1 has been implicated in many pathways, including cell morphology and cell cycle regulation. Slf1 and Pom1 work with Skb1 in the morphology and cell cycle regulation, respectively. To ask whether Skb1 promote GCRs through these pathways, we determined the effect of *slf1* Δ or *pom1* Δ on GCRs (Fig. 6a) and found that neither *slf1* Δ nor *pom1* Δ reduced GCRs, showing that Skb1 promotes isochromosome formation independently of Slf1 or Pom1.

Line 253-263. "Skb1 has been implicated ... function independent of Slf1 or Pom1."

Line 351-356. "Skb1 has been implicated in ... did not reduce GCR rates."

We discussed that both Skb1 and PCNA K107 promote Rad52-dependent GCRs but not DNA repair, suggesting the functional link between Skb1 and PCNA.
Line 356-361. "In *rad52Δ* cells, ... Rad52-dependent GCRs."

Reviewer #3 (Remarks to the Author):

Mongia et al : Fission yeast Srr1 and Skb1 promote isochromosome formation at the centromere.

Here the authors have sought factors that cause GCRs in a *rad51Δ* mutant strain and find Srr1 (Ber1) and Skb1 (the orthologue of PRMT5). Using a previously described minichromosome ChLC they show that these factors promote isochromosome formation at centromeres and that mutations in these factors, including mutations in the arginine methyltransferase domain of SKb1 lead to elevated levels of mutations in a *rad51Δ* background. Further analysis reveals Srr1 to be required for the DNA damage response and to promote DNA damage independently of Rad51, and to have both overlapping and non-overlapping roles with Rad52 in GCR formation. Further the SRR1-like domain is required for this function in SRR1, while a beautiful structural analysis suggests that the Skb1 arginine methyltransferase acts in the Rad52-dependent GCR pathway.

The manuscript is very well and clearly written, and the data they present are original, very clear and largely support the conclusions drawn. However, the data, while interesting, remain somewhat preliminary and raise more questions than they really answer. For example, there are no mechanistic insights into how these two mutations limit GCRs in the various backgrounds; whether they limit GCRs through impacting the initiation of these spontaneous GCR-inducing events, or whether they are required in the processing mechanisms leading to GCR formation? Further, while these data suggest that Skb1 functions in the Rad52 GCR pathway, it is perhaps surprising that loss of Skb1 does not show any increased sensitivity to any of the DNA damaging agents tested, which *rad52Δ* clearly does.

srr1Δ reduced GCRs and increased DNA damage sensitivity. To know whether Srr1 is required for DNA damage checkpoint response, we performed additional experiments and found that Srr1 is not essential for MMS-induced cell cycle arrest (Fig. 3b) or for Chk1 phosphorylation (Fig. 3c). These data suggest that Srr1 promotes DNA repair process leading to GCRs.

Line 176-190. "Cell cycle arrest caused ... the checkpoint activation."

Line 309-312. "Srr1 was not essential ... do not promote GCRs."

We performed additional experiments and found that *srr1Δ* increased the fraction of cells containing Rpa2-mCherry foci (Fig. 4d), suggesting that *srr1Δ* accumulates single-stranded DNA.

Line 229-236. "Given the hypersensitivity of *srr1Δ* ... single-stranded DNA in *srr1Δ* cells."

Line 317-321. "*srr1Δ* caused synthetic growth ... spontaneous DNA damage."

Skb1 has been implicated in many pathways, including cell morphology and cell cycle regulation. Slf1 and Pom1 work with Skb1 in the morphology and cell cycle regulation, respectively. To ask whether Skb1 promotes isochromosome formation through these pathways, we determined the effect of *slf1Δ* or *pom1Δ* on GCRs (Fig. 6a) and found that neither *slf1Δ* nor *pom1Δ* reduced GCR rates in *rad51Δ* cells, showing that Skb1 promotes isochromosome formation independently of Slf1 or Pom1.

Line 253-263. "Skb1 has been implicated ... function independent of Slf1 or Pom1."

Line 351-356. "Skb1 has been implicated in ... did not reduce GCR rates."

We also discuss the similarity of *pcn1-K107R* and *skb1* mutant phenotypes and the possibility that, in concert with PCNA, Skb1 forms DNA structures leading to Rad52-dependent GCRs.

Line 356-361. "In *rad52*Δ cells, ... Rad52-dependent GCRs."

Lastly, it is not entirely clear why both of these mutations are included together in the same manuscript, as clearly the double mutant has increased sensitivity suggesting non-overlapping functions leading to GCR's. In summary, the story is novel and encouraging but would benefit from further mechanistic insights.

We think it is worth showing both *Srr1* and *Skb1* because they are identified in the same clone in the screening and are the first examples showing additive effects on isochromosome formation. *skb1-A377V* did not significantly reduce GCRs but reduced GCRs in the presence of *srr1-W157R* (Fig. 1e), suggesting a functional relationship between them.

We believe that additional experimental data and analyses we performed in response to the referees' comments to gain mechanistic insights have improved this paper.

Reviewers' comments:

Reviewer #1 (Remarks to the Author):

The authors have satisfactorily responded to my inquiries. I recommend publication of this work.

Reviewer #2 (Remarks to the Author):

The authors have added some additional, but limited data to their manuscript. The primary concern that this is really not providing a mechanism for either of these suppressors is still there. The Srr1 pathway seems to be most developed, but it still is largely phenomenological. They still fail to define growth features of the double or single mutants, cellular morphology, growth rate, etc.

Specific comments:

They did not verify that the rad52 mutants used here do not have the fbh1 suppressor (referring to their earlier work does not obviate this concern; rad52 notoriously picks up this suppressor). The absence of fbh1 effect in rad51 is not relevant.

Srr1/Ber1: the standard name in pombase is Ber1. The deletion and other mutants appear largely uncharacterized.... and remains so in this paper. If they are reporting the first characterization of these mutants, they really do need to describe them properly. That includes cell growth dynamics, cell morphology plus/minus drugs, etc.

Skb1 interacts with a wide range of proteins outside of the cell morphology pathway, including (according to pombase), proteins affecting the cell cycle, chromatin modification, and PAK kinase. It's not a surprise the cell morphology pathway isn't required for this phenotype.

Finally, the method used to generate the mutations is not well described in this or their previous paper. I can find no reference for the use of sodium nitrate as a mutagen, or the rate of mutations that occurred.

Reviewer #4 (Remarks to the Author):

In this revised manuscript, the authors have addressed the concerns raised in previous review. The manuscript is supported by more robust data, along with key new experiments that directly evaluate the role of Srr1 and Skb1 in promoting isochromosome formation at the centromere.

This is an interesting and carefully executed study with conclusions supported by the data, and should be of broad interest to the Communications Biology community.

Point-by-point Response to Reviewers' comments

Thank you for the time to review our manuscript. We appreciate insightful comments from the reviewers. We highlighted all the changes we made in the manuscript text file.

Reviewers' comments:

Reviewer #1 (Remarks to the Author):

The authors have satisfactorily responded to my inquiries. I recommend publication of this work.

I appreciate your support.

Reviewer #2 (Remarks to the Author):

The authors have added some additional, but limited data to their manuscript. The primary concern that this is really not providing a mechanism for either of these suppressors is still there. The *Srr1* pathway seems to be most developed, but it still is largely phenomenological. They still fail to define growth features of the double or single mutants, cellular morphology, growth rate, etc.

Specific comments:

1. They did not verify that the *rad52* mutants used here do not have the *fbh1* suppressor (referring to their earlier work does not obviate this concern; *rad52* notoriously picks up this suppressor). The absence of *fbh1* effect in *rad51* is not relevant.

As the reviewer suggested, we additionally sequenced the *fbh1* gene in all the *rad52* mutant strains used in this work and confirmed no *fbh1* mutations (Supplementary Data 2).

Line 355-357:

"We also confirmed no *fbh1* mutations in all the *rad52* mutant strains and the *srr1*Δ *rad51*Δ strain used in this study (Supplementary Data 2)."

We also explained in the Methods that we did not pick up exceptionally large colonies of *rad52* mutants.

Line 408-409:

“We did not pick up exceptionally large colonies of *rad52* mutants because they can contain an *fbh1* mutation⁶².”

2. *Srr1/Ber1*: the standard name in pombase in *Ber1*. The deletion and other mutants appear largely uncharacterized.... and remains so in this paper. If they are reporting the first characterization of these mutants, they really do need to describe them properly. That includes cell growth dynamics, cell morphology plus/minus drugs, etc.

As the reviewer suggested, we determined the doubling time for wild-type and *srr1Δ* cells (Table E in Supplementary Data 1) and found that *srr1Δ* prolonged the doubling time.

Line 183-185:

“The doubling time of wild-type and *srr1Δ* cells grown in EMM media at 30°C were 2.48 ± 0.11 and 2.73 ± 0.10 h, respectively ($p = 0.042$, the two-tailed student’s *t*-test) (Table E in Supplementary Data 1).”

Exp-1 (h)	wild type (x10E5)	wild type log2	srr1Δ (x10E5)	srr1Δ log2
0	37.2	5.22	41.2	5.36
1	43.7	5.45	45.7	5.51
2	56.5	5.82	57.6	5.85
3	78.6	6.30	77.1	6.27
4	102	6.67	98.6	6.62
5	125	6.97	114	6.83
6	166	7.38	149	7.22
7	227	7.83	199	7.64
8	273	8.09	262	8.03
9	423	8.72	351	8.46
10	559	9.13	477	8.90
slope		0.393		0.358
doubling time		2.54		2.79

Exp-2 (h)	wild type (x10E5)	wild type log2	srr1Δ (x10E5)	srr1Δ log2
0	41.3	5.37	40.7	5.35
1	45.2	5.50	44.8	5.49
2	63.4	5.99	58.1	5.86
3	75.3	6.23	71.3	6.16
4	132	7.04	119.0	6.89
5	178	7.48	142	7.15
6	232	7.86	176	7.46
7	272	8.09	221	7.79
8	401	8.65	320	8.32
9	523	9.03	409	8.68
10	666	9.38	498	8.96
slope		0.424		0.382
doubling time		2.36		2.62

Exp-3 (h)	wild type (x10E5)	wild type log2	srr1Δ (x10E5)	srr1Δ log2
0	35.1	5.13	33.6	5.07
1	45.0	5.49	43.0	5.43
2	58.1	5.86	55.0	5.78
3	79.3	6.31	76.1	6.25
4	108	6.75	93.4	6.55
5	131	7.03	115	6.85
6	176	7.46	149	7.22
7	237	7.89	187	7.55
8	306	8.26	254	7.99
9	408	8.67	322	8.33
10	560	9.13	467	8.87
slope		0.393		0.358
doubling time		2.54		2.79

Doubling time		
	wild type	srr1Δ
Exp-1	2.54	2.79
Exp-2	2.36	2.62
Exp-3	2.54	2.79
mean	2.48	2.73
s.d.	0.11	0.10
T-test		0.042

We also created a new figure (Supplementary Fig. 2a), showing the images of cell morphology of wild-type, *chk1* Δ , *srr1* Δ , *srr1-W157R*, *srr1-H148A*, and *srr1-D111A,P112A* strains either in the presence or absence of MMS and explain that, like wild-type cells, *srr1* Δ cells were elongated after MMS exposure in the Results.

Line 185-187:

“Like wild-type cells, *srr1* Δ cells were elongated after MMS exposure (Supplementary Fig. 2a). These data suggest that Srr1 is dispensable for DNA damage-induced cell cycle arrest.”

Supplementary Fig 2

Supplementary Fig. 2 a Cell morphology of wild-type, *chk1* Δ , *srr1* Δ , *srr1-W157R*, *srr1-H148A*, and *srr1-D111A,P112A* strains (TNF35, 3559, 5943, 9011, 9007, and 8990). Early log-phase EMM cultures were divided into two aliquots and incubated for 8 h at 30°C either in the presence (bottom row) or absence (top row) of 0.01% MMS. DIC, differential interference contrast. A bar indicates 10 μ m. Cells were observed using a DeltaVision Personal fluorescence microscopy system (see Methods). **b** Colony formation of wild-type, *srr1* Δ , *srr1-W157R*, *srr1-H148A*, and *srr1-D111A,P112A* strains (TNF3885, 5837, 8280, 8275, and 8274) on YE3S plates. Note that, compared to wild-type cells, *srr1* Δ , *srr1-W157R*, *srr1-H148A*, and *srr1-D111A,P112A* strains formed small colonies, especially at 36°C.

We also created a new figure (Supplementary Fig. 2b), showing colonies of wild-type, *srr1* Δ , *srr1-W157R*, *srr1-H148A*, and *srr1-D111A,P112A* strains and explain that, like *srr1* deletion, the mutations in the SRR1-like domain reduced the colony size in the Results and Discussion.

Line 252-256:

“As expected from the prolonged doubling time (Table E in Supplementary Data 1), *srr1* Δ cells formed small colonies on plate media compared to wild-type cells. Like *srr1* Δ cells, *srr1-W157R*, *srr1-D111A,P112A*, and *srr1-H184A* cells produced small colonies (Supplementary Fig. 2b), consistent with the role of the SRR1-like domain even in the absence of exogenous DNA damage.”

Line 326-327:

“*srr1* Δ increased the doubling time and produced small colonies compared to wild type (Table E in Supplementary Data 1; Supplementary Fig. 2b).”

3. Skb1 interacts with a wide range of proteins outside of the cell morphology pathway, including (according to pombase), proteins affecting the cell cycle, chromatin modification, and PAK kinase. It's not a surprise the cell morphology pathway isn't required for this phenotype.

Yes, I added Shk1 (PAK) to the list of Skb1 interacting proteins and discussed how Skb1 might cause isochromosome formation.

Line 375-382:

“Interestingly, the proteins affecting chromatin structure and replication such as histones, Fen1 DNA-flap endonuclease, and p53 have been found as PRMT5 substrates^{65,66,67,68}. Skb1 also interacts with Shk1, a p21-activated kinase (PAK), that negatively regulates cell cycle progression⁶⁹. Mutating the residues essential for the RMTase activity⁵⁷: *skb1-F319Y* and *skb1-E422A,E431A* strongly reduced GCR rates, suggesting that Skb1 promotes GCRs through the RMTase activity. It is important to identify the key substrate(s) of Skb1 RMTase that induces chromosome instability in the future.”

4. Finally, the method used to generate the mutations is not well described in this or their previous paper. I can find no reference for the use of sodium nitrate as a mutagen, or the rate of mutations that occurred.

Thank you for the careful reading of this manuscript. As the reviewer suggested, we included a reference paper to the nitrous acid/sodium nitrate mutagenesis (Prakash L. Genetics 1974) and explain more details on how we introduced random mutations into *rad51* Δ cells.

Line 410-430:

“To search for the genes causing GCRs in *rad51* Δ cells, we introduced random mutations into yeast essentially, as described previously³². Nitrous acid was used as the mutagen because it efficiently introduces mutations in DNA-repair deficient cells as in wild-type cells⁷². *rad51* Δ cells containing ChL^C (TNF5411) grown in EMM were collected at the log phase (5×10^6 cells mL⁻¹), suspended in water, and kept overnight at 4°C. After centrifugation, cells were suspended in 0.8 mL of 0.01 M nitrous acid solution prepared before use by dissolving sodium nitrate (Wako,195-20562) in 0.5 M sodium acetate (pH

4.8), and incubated at room temperature for 20 min. After adding an equal volume of the stop buffer (3.6% Na₂HPO₄·12H₂O and 1% yeast extract) to the cell suspension, cells were plated on EMM+UA plates. The plating efficiency of the mutagenized cells determined using EMM plates was around 10%. 24,000 independent clones were incubated as patches on EMM+UA plates for 2–3 d at 30°C, and then transferred onto 5-FOA+UA plates to semi-quantitatively determine the rate of uracil auxotroph, resulting from GCR or a point mutation in the *ura4* gene. 80 clones produced reduced numbers of colonies on 5-FOA+UA plates. PFGE analysis showed that six of them contained the aberrant sizes of the parental ChL^C. The remaining 74 clones were crossed with wild-type cells containing ChL^C. Three clones reproducibly exhibited reduced GCR rates. Deep sequencing of genomic DNA was carried out using MiSeq (Illumina, San Diego, CA), and the mutations were identified by pooled-linkage analysis^{73,74}. Nucleotide sequence data of the parental strain and a pool of nine mutant segregants obtained by backcrossing one of the three clones are available in the DDBJ Sequenced Read Archive under the accession numbers DRX042095 and DRX042098, respectively.”

Reviewer #4 (Remarks to the Author):

In this revised manuscript, the authors have addressed the concerns raised in previous review. The manuscript is supported by more robust data, along with key new experiments that directly evaluate the role of Srr1 and Skb1 in promoting isochromosome formation at the centromere.

This is an interesting and carefully executed study with conclusions supported by the data, and should be of broad interest to the Communications Biology community.

Thank you for your support.